

Atmospheric
Measurement
Techniques

# Real-world measurement and mechanical-analysis-based verification of NO$_x$ and CO$_2$ emissions from an in-use heavy-duty vehicle CE1

**Hiroo Hata**[1], **Kazuo Kokuryo**[2], **Takehiko Ogata**[2], **Masahiko Kugata**[1], **Koichi Yanai**[1], **Megumi Okada**[1], **Chikage Funakubo**[1], **Minoru Yamazaki**[1], and **Junya Hoshi**[1]

[1]Tokyo Metropolitan Research Institute for Environmental Protection, 1-7-5 Sinsuna, Koto-ku, Tokyo 136-0075, Japan
[2]Modern Planning Inc., 5-49-10 Chuo, Nakano-ku, Tokyo 164-0011, Japan

**Correspondence:** Hiroo Hata (hata-h@tokyokankyo.jp, hata-hiroo@aist.go.jp)

**Abstract.** A portable emission measurement system (PEMS) was used to measure the real-world driving emissions pertaining to a Japanese middle-sized heavy-duty vehicle. The testing was performed with the vehicle being driven in the metropolitan area of Tokyo in four seasons (January, June, August, and November) to analyze the seasonal dependence of NO$_x$ and CO$_2$ emissions. The experimental results indicated that the amount of NO$_x$ emissions was particularly high in the cold season owing to the slow starting of the NO$_x$ after-treatment systems, which is to say the exhaust gas recirculation and urea-selective-catalytic-reduction systems, under low-ambient-temperature conditions. In real-world driving, a high acceleration pattern was observed in the low-speed region which is not considered in the world harmonized vehicle cycle, which is the worldwide official driving mode in the chassis dynamometer experiment. Finally, the transient emission tables for NO$_x$ and CO$_2$ were constructed based on the PEMS measurement results and the classical mechanic theory. The constructed tables replicated well the experimental results in all the considered conditions involving different ambient temperatures and locations. The proposed approach can be used to evaluate emission inventories in the future.

# 1 Introduction

The air pollution caused by long-term air pollutants, which are chemically stable components such as CO$_2$, and short-term air pollutants, which are reactive chemicals such as NO$_x$, volatile organic compounds (VOCs), and photochemical oxidants, is a cause of significant concern in many countries, including the United States, the European Union, China, India, and Japan (Costa et al., 2014; Akimoto et al., 2015; Ravindra et al., 2016; Yang and Wang, 2017; Sullivan et al., 2018). CO$_2$, which is a greenhouse gas and contributes to climate change, is a critical long-term air pollutant. According to the worldwide observation data, the global temperature has increased by approximately 1 °C by 2020 from the average temperature from 1951 to 1980 (National Aeronautics and Space Administration, 2020 TS1). Climate change continues to occur, with the global temperature increasing annually. It has been reported that if strategies to eliminate greenhouse gas emissions from anthropogenic sources are not adopted, the global temperature will increase by more than 1.5 °C (Saito, 2010; Intergovernmental Panel on Climate Change, IPCC, 2018). In addition, photochemical oxidants mainly composed of ozone are well-known short-term air pollutants generated by the photochemical reaction of NO$_x$ and VOCs (Sillman, 1999). The concentration of photochemical oxidants in the atmosphere is a significant concern for humans, animals, and crops in many countries (Chappelka and Samuelson, 1998; O'Neill et al., 2004; Wang et al., 2017). The Japanese government has set the environmental standard for photochemical oxidants as 0.06 ppm (parts per

million) per hour; however, the values in all the tested sites in 2017 exceeded this standard value (Ministry of the Environment 1 TS2). To address the problems of climate change and photochemical pollutants, it is necessary to mitigate the emissions of long- and short-term air pollutants. In this context, heavy-duty vehicles represent a major emission source of $CO_2$ and $NO_x$ (Seo et al., 2016, 2018; Hata and Tonokura, 2019). According to statistical reports, almost 20 % of the total $CO_2$ and $NO_x$ emissions in Japan are produced by heavy-duty vehicles (Ministry of the Environment 2), and thus, it is necessary to implement laws to reduce the exhaust emissions from heavy-duty vehicles. However, prior to setting new regulatory limits for exhaust emissions, experimental testing must be performed to understand the emission trend corresponding to the target vehicles. Until recently, the vehicle exhaust emissions were experimentally examined by conducting laboratory tests using a chassis dynamometer. The advantage of such testing is the high repeatability of each test; moreover, several types of vehicles can be tested under nearly identical conditions, such as the driving mode and environmental temperature. However, in the chassis dynamometer experiment, changes in the seasonal ambient temperature and road gradient are not considered. In general, the laboratory temperature is set at approximately 25 °C, and it cannot be easily set to an arbitrary temperature (especially a low temperature range) owing to the exhaust heat from vehicle and measurement machines via the normal laboratory system (recently, however, temperature-variable chassis dynamometers were adopted only in a limited number of laboratories; Clairotte et al., 2013; Ko et al., 2017). It has been noted that environmental temperature considerably influences the amount of exhaust emissions (detailed explanations regarding this observation are provided in Sect. 3.1), leading to the release of a large amount of pollutants (including $NO_x$) into the atmosphere in low-ambient-temperature conditions (the MOVES2010 Report by the U.S. Environmental Protection Agency, 2010). Moreover, the road gradient also influences the amount of exhaust emissions because it directly affects the driving force, which is presumed to have a negative effect on fuel consumption (or $CO_2$ emissions) and other exhaust emissions. Road temperature and gradient might vary from one season and location to another; therefore, real driving emission measurements are more suitable for a better understanding of the real-world driving emission (RDE) verification. For this reason, the European Union has implemented a regulation for RDEs from light-duty vehicles using portable emission measurement systems (PEMS) (Valverde et al., 2020). Consequently, countries such as the United States, China, and India have made decisions to implement RDE measurements as a regulatory test, and Japan is now considering the implementation of regulations via real-world measurements using PEMS (Giechaskiel et al., 2019). Currently, in terms of the research field of atmospheric science, many researchers have conducted RDE measurements using PEMS. Gallus et al. (2017) measured the exhaust emissions

from two diesel light-duty vehicles using PEMS and analyzed the relationship between the emission trends of $CO_2$ and $NO_x$ and road properties such as the road gradient and driving style. It was noted that the road gradient linearly affected the amount of $CO_2$ and $NO_x$ emissions, thereby indicating the importance of obtaining measurements from the vehicles in road tests. Mendoza-Villafuerte et al. (2017) measured the emission trend from a heavy-duty vehicle by using PEMS and developed an analysis method based on the geographic information system. The results clarified the notable influence of the road boundary condition, land-use data, and speed-limit data on the amount of emissions. The accuracy of PEMS is also important in discussing RDE test results. Cao et al. (2016) conducted a study to clarify the accuracy of gaseous pollutants measured by PEMS, concluding that $NO_x$ emissions would sometimes be overestimated in the low $NO_x$ concentration range, accounting for a $\sim 50\%$ maximum from the reference $NO_x$ concentration owing to analyzer drift. These results suggest the limitation of PEMS' accuracy when the measurement system is used in this type of approval test. Moreover, several studies were conducted using PEMS to measure the basic data from both light-duty and heavy-duty vehicles (Liu et al., 2009; Kousoulidou et al., 2013; O'Driscoll et al., 2016; Kwon et al., 2017; Luján et al., 2018). Nevertheless, the conduction of road measurement experiments using PEMS is a relatively new domain compared with laboratory tests, including chassis dynamometer experiments, and only a few studies have been performed to assess the analytical data, for instance, in terms of the mathematical formulation of the effect of ambient temperature on the exhaust emission and application of the measurement results to evaluate the emission inventory.

The purpose of this study is two-fold. First, chassis-dynamometer-based and RDE measurements using PEMS were conducted on a heavy-duty Japanese vehicle to determine the importance of RDE-specific factors, including the ambient temperature and road gradient, among others. Second, the obtained experimental results were analyzed based on two parameters, i.e., the driving force and vehicle speed, to develop an analytical method to evaluate the amount of $CO_2$ and $NO_x$ emissions from the vehicle in an arbitrary driving condition. Moreover, the difference between chassis dynamometer and PEMS results was quantified to understand the difference between RDE and laboratory measurement results. To the best of our knowledge, this is the first study in which PEMS measurement results are applied in the development of an estimation model for vehicular exhaust. It was expected that the proposed method could be used to quantify the results of RDE measurements, and the analyzed data could be applied to evaluate the emission inventory from vehicles in the future.

## 2 Methodology

Three methods were used in this study: laboratory tests using the chassis dynamometer, RDE measurements using PEMS, and data analysis to construct the estimation model for the emissions inventory. For a detailed analysis of the experimental results from the RDE measurements, data processing methods included data smoothing for the high-resolution time profiles of the emissions data and the extraction of the road gradient from official open sources. Details on these methods are described in Sect. 2.1 to 2.3.

### 2.1 Laboratory test using chassis dynamometer

The exhaust emission from a diesel heavy-duty vehicle used in the Japanese market was measured using a chassis dynamometer (MEIDACS-DY6200P, Meiden Engineering Co.). The purpose of the laboratory tests was to determine the difference between chassis dynamometer measurements and RDE measurement results. The tests were also performed to verify whether the conditions under which the vehicles performed were the same in the four different seasons that were investigated. This enabled a comparison of the seasonal dependencies of real-world measurements. The measured vehicle was equipped with a diesel particulate filter and urea selective catalytic reduction (urea-SCR) system, and it met the current Japanese regulation set in 2016, of which details are described in Table S1 (Supplement). The specifications of the vehicle are listed in Table 1. The emission components including CO, $CO_2$, NO, $NO_2$, $CH_4$, total hydrocarbon (THC), and $N_2O$ were measured using the vehicle's emission measurement system (MEXA-7400D, HORIBA Ltd.). In particular, the amounts of CO, $CO_2$, $CH_4$, and $N_2O$ were measured using a nondispersive infrared sensor. The NO and $NO_2$ amounts were measured using the chemiluminescence method, and the amount of THC was measured using a flame ionization detector (FID). The detailed composition of the THC, including the VOCs, was measured using a gas chromatograph mass spectrometer and FID (GC-MS/FID; GCMS-QP2020, Shimadzu Corp.) and liquid chromatograph mass spectrometer (LC-MS; G6120B Quadrupole LC/MS, Agilent Technologies Inc.). The details of the VOC analysis method have been provided in a previous study (Hata et al., 2019). Although the VOC analysis is beyond the scope of this study, the information may be useful for researchers who aim to examine the composition of the VOCs emitted from heavy-duty vehicles; therefore, the analyzed data are provided in the Supplement. The tested driving mode was the world harmonized vehicle cycle (WHVC) with cold and hot starts. Laboratory tests were conducted in all the seasons; however, VOC analyses were performed in only three seasons, which is to say in November 2018, June 2019, and August 2019. In all the laboratory measurements, room temperature was set to be approximately 25 °C.

**Table 1.** Specifications of the vehicle tested in this study (DPF signifies diesel particulate filter)CE3.

| Category | Heavy-duty |
|---|---|
| Regulation year | 2016 |
| Fuel | Diesel |
| Displacement (L) | 5193 TS3 |
| Vehicle weight (kg) | 4920 |
| Detoxification tool | EGR, DPF, urea-SCR |
| Official fuel consumption (L/km) | 0.128 |
| Total driving distance (km) | 27 241 |

### 2.2 Real-world measurement using PEMS

A PEMS (OBS-ONE, HORIBA Ltd.) was used to perform the road emission measurement. The measured components included CO, $CO_2$, NO, $NO_2$, and THC, and the measurement techniques were the same as those in the chassis dynamometer experiment, as described in Sect. 2.1. The tests were conducted in four seasons to investigate the seasonal dependence of the emissions: autumn (19–21 November 2018), winter (15–17 January 2019), spring (10–14 June 2019), and summer (26–30 August 2019). All the tests were conducted twice for each day: in the morning and afternoon. The vehicle speed, ambient temperature, humidity, exhaust gas recirculation (EGR) ratio, urea-SCR temperature, and urea injection ratio were determined using the information of an onboard device (OBD) in the vehicle. The EGR ratio was measured only in the spring and summer tests. The driving route was the same across all days and seasons. The driving route included the city and bay area of Tokyo, with the total driving distance being 28.5 km.

### 2.3 Experimental data processing

#### 2.3.1 Data smoothing of vehicle speed and acceleration and time delay treatment between observed concentration and vehicle acceleration

The vehicle speed (km/s)TS4 was monitored in the experimental process at 10 Hz. The vehicle acceleration (km/s$^2$)TS5 was calculated by determining the differential of the vehicle speed. Because the dispersion of the acceleration calculated using the data of the vehicle speed obtained at 10 Hz appeared to be large, the vehicle speed data in arbitrary time, $v_t$, was smoothed using Eq. (1).

$$v_t = \sum_{i=t-2}^{t+2} \frac{v_i}{5} \tag{1}$$

In Eq. (1), vehicle speed was smoothed using the five-point average of the speeds at neighboring time steps. We determined the averaging number of vehicle speed as five points based on two concepts. First, the averaging number of vehicle speed should be minimized as much as possible to main-

*Please note the remarks at the end of the manuscript.*

tain the high-resolution time step. Second, the dispersion should be sufficiently lower than the measured vehicle speed. It is also worth noting that the averaging number of vehicle speed may depend on the measurement tools, and the five-point value was suitable in this study. Using the smoothed vehicle speed, the acceleration in arbitrary time, $a_t$, was calculated using the central difference method, as follows:

$$a_t \cong \frac{v_{t+1} - v_{t-1}}{2\Delta t}, \tag{2}$$

where $\Delta t$ is the time step of the monitoring time duration ($= 0.1$ s).

The exhaust emission was measured using the analyzer installed in the chassis dynamometer or PEMS, and the OBD information was collected directly from the vehicle. Owing to the different approaches employed, the measured exhaust emissions and velocity (or acceleration) involved a time delay. To compensate for this delay, a statistical method was applied. As discussed in the subsequent section, the target pollutants, NO$_x$ and CO$_2$, increased when the vehicle accelerated, indicating that the concentration of the pollutants and acceleration are correlated. The cross-correlation function between two parameters $x$ and $y$ at time $\tau$, $C_{xy}(\tau)$, can be defined as follows:

$$C_{xy}(\tau) = \frac{R_{xy}(\tau)}{\sqrt{R_{xx}(0) R_{yy}(0)}}, \tag{3}$$

where $R_{xx}(0)$ and $R_{yy}(0)$ are the autocorrelation function of $x$ and $y$ at the base time, respectively, and $R_{xy}(\tau)$ is the cross-correlation function at time $\tau$. The time $\tau$ corresponding to the maximum cross-correlation function (defined in Eq. 3) was numerically determined and defined as the time delay between the emission peak and acceleration.

### 2.3.2 Evaluation of driving force

The vehicle driving force, $F$, can be defined as follows:

$$F = (m + \Delta m)a + \mu_r mg + mg\sin\theta + \mu_a A v^2, \tag{4}$$

where $F$, $m$, $\Delta m$, $a$, $\mu_r$, $g$, $\theta$, $\mu_a$, $A$, and $v$ denote the driving force (N), weight of inertia of the vehicle (kg), weight of the rotatory parts of the vehicle (kg), vehicle acceleration (m/s$^2$), rotation friction coefficient, gravity due to acceleration (9.8 m/s$^2$), slope angle, air friction coefficient, area of the front side of the vehicle (m$^2$), and vehicle speed (m/s), respectively. While the coast down test to determine $\mu_r$ and $\mu_a$ was not conducted in this study, based on our previous study, $\mu_r$ and $\mu_a$ were set at 0.0089 and 0.0027, respectively, given that we used the same type of heavy-duty vehicle that was tested in our previous study. The threshold of acceleration was defined as 0.139 m/s$^2$ ($= 0.5$ km/(h s)), and if the acceleration was less than this value, the acceleration was set at zero (Yoshizumi et al., 1980). The total

vehicle weight, $m$, including the weight of the vehicle itself (4920 kg), cargo such as PEMS ($\simeq$ 200 kg) and four batteries for PEMS (37 kg $\times$ 4 $\simeq$ 150 kg), driver and operator (55 kg $\times$ 2 $\simeq$ 110 kg), and other measurement-related parts (500 kg), was set at 5880 kg. The parameter $\Delta m$, pertaining to the weight of the transmission system and tires, was set at 0.10 (during acceleration) and 0.07 (in the case of constant speed), respectively. $A$ was set at 7.5725 m$^2$, as reported in a tutorial of the tested vehicle. The $\theta$ value was extracted from the altitude information derived from the aviation laser surveying data (ALSD; Geospatial Information Authority of Japan, 2020 TS6). In particular, the ALSD includes the altitude information in an arbitrary area in Japan with a mesh size of less than 2 m $\times$ 2 m. The data assessment was performed by conducting the following three steps.

- Two types of ALSD are available: original and filtered data. The original data include the complete information of the ALSD, whereas the filtered data include the ALSD information with the buildings and trees filtered to ensure that the users can assess the usable land information. The filtered ALSD were selected in this study.

- The altitude data from the ALSD were sorted into 1 m grids, and the altitude of each mesh was set considering the nearest altitude in the ALSD. These altitudes were smoothed using the mean average in the vicinity of $\sim$ 5 m meshes. This smoothed value in the vicinity of $\sim$ 5 m meshes was determined by varying the averaging mesh number (e.g., from 1, 2 . . . ) while carefully checking to ensure that the noise remained negligible relative to the height change.

- Using the determined altitude, the road slope was calculated by considering a tangent of $\sim$ 7 m meshes in the vicinity of two meshes. This smoothed value, in the vicinity of $\sim$ 7 m meshes, was determined using the same method that was applied to altitude data, as described above.

## 3 Results and discussion

### 3.1 Seasonal trend of measured NO$_x$ emissions in real-world driving

The time profile of the NO$_x$ emissions in four seasons is shown in Fig. 1. It can be noted that the NO$_x$ emissions in real-world driving are season-dependent and inversely proportional to the ambient temperature. According to Fig. 1, the time profile of the NO$_x$ emissions can be divided into three phases: high-emission phase from the start of the driving time to 10 min (phase 1), medium- to low-emission phase in the driving time from 10 to 30 min (phase 2), and low-emission phase in the driving time after 30 min (phase 3). These three phases are related to the operation of the EGR and urea-SCR. In general, the EGR system decreases the O$_2$ concentration

**Atmos. Meas. Tech., 14, 1–12, 2021**                **https://doi.org/10.5194/amt-14-1-2021**

in the intake air to reduce the amount of $NO_x$ generated inside the combustion chamber by recirculating the exhaust gas to the intake air (Abd-Alla, 2002). The urea-SCR system reduces the $NO_x$ concentration in the exhaust gas through the redox reaction between $NO_x$ and $NH_3$, which is produced by the hydrolysis of urea (Fang and DaCosta, 2003; Upadhyay and Van Nieuwstadt, 2006; Hsieh and Wang, 2011). The operation of the EGR and urea-SCR systems is usually avoided in the initial stage of driving because in the cold-start process, the exhaust gas temperature and catalysis surface temperature are low, likely leading to the deposition of the particulate matter in the EGR system and urea leakage from the SCR system. Moreover, the operation of the EGR is avoided in the cold-start phase to prevent the occurrence of accidental fires in the combustion chamber in the presence of unburned fuel. Figure S1 shows the relationship between the EGR ratio (%) and engine coolant temperature of the tested vehicle. Figure S1 suggests that in the cold-start situation, the engine coolant temperature was proportional to the exhaust gas temperature. Further, in general, engine coolant temperature is usually used to control the EGR system. Figure S1 shows that the EGR begins to operate when the engine coolant temperature reaches 60 °C. The time profile of the engine coolant temperature measured in the four seasons is shown in Fig. S2. The dependence of the coolant temperature on the season (or the ambient temperature) is notable. The coolant temperature increases to 60 °C more rapidly in the hotter seasons than in the colder seasons because the initial coolant temperature is higher in the hotter seasons. Therefore, the EGR system starts operating earlier in hot seasons such as summer; consequently, the amount of $NO_x$ emissions in hot seasons is lower than that in the cold seasons. Figure S3 shows the relationship between the urea injection amount and SCR surface temperature. It can be noted that the urea injection started after the SCR temperature became 150 °C. The time profile of the SCR temperature in real-world driving in four seasons is illustrated in Fig. S4. Similar to the trend in the engine coolant temperature, the SCR temperature increases to 150 °C more rapidly in hot seasons than in cold seasons. Therefore, the season (or ambient temperature) influences the effectiveness of the urea-SCR system, and the $NO_x$ emissions are thus lower in hot seasons such as summer. According to Fig. 1, phase 1 corresponds to the period in which both the EGR and urea-SCR systems are not operating. In phase 2, the EGR system is operating, and the $NO_x$ emissions are thus partly purified. In phase 3, both the EGR and urea-SCR systems are operating, resulting in a high detoxification of $NO_x$. These three phases considerably depend on the ambient temperatures, and colder seasons, such as winter, involve a higher amount of $NO_x$ emissions. Figure 2a shows the relationship between the ambient temperature and total $NO_x$ emissions per driving distance (g/km). According to Fig. 2a, the $NO_x$ emissions are dominant in phase 1. The magnitude of the emissions is 2 to 7 times lower in phase 2 than that in phase 1, and nearly no emissions occur in phase 3. Thus, the

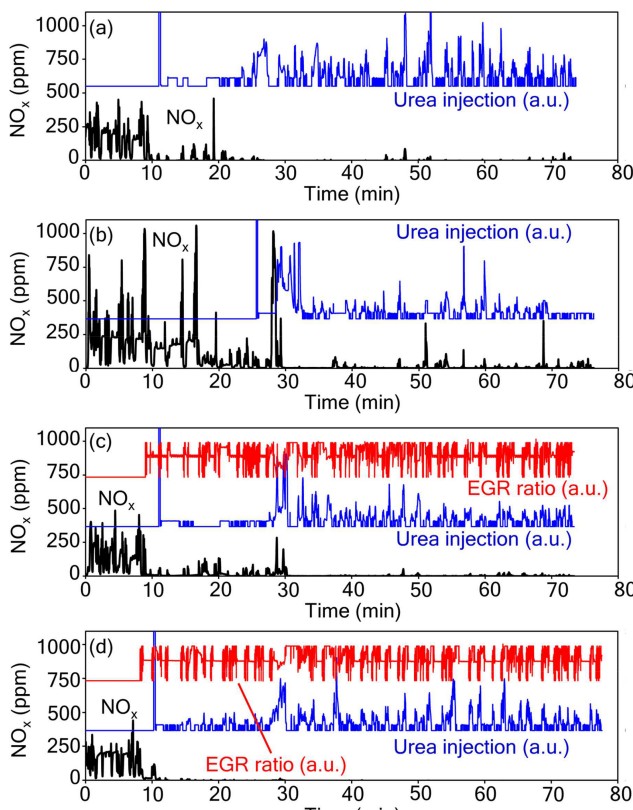

**Figure 1.** Time profiles of measured $NO_x$ emission and driving distance in four seasons: **(a)** autumn (morning test results from 19 November 2018), **(b)** winter (morning test results from 15 January 2019), **(c)** spring (morning test results from 12 June 2019), and **(d)** summer (morning test results from 28 August 2019). The term a.u. signifies arbitrary unit. CE4

cold-start emission in phase 1 is a critical phase to mitigate the amount of $NO_x$ emissions in the atmosphere.

## 3.2 Measured $CO_2$ emissions in real-world driving

Figure 2b shows the relationship between the ambient temperature and total $CO_2$ emissions per distance (g/km). The temperature dependence of the $CO_2$ emissions is lower than that of the $NO_x$ emissions. In some cases, after the EGR started operating, the $CO_2$ emissions were high; however, this phenomenon likely occurred owing to the error in the driving pattern in each real-world driving test. Moreover, although the amount of $CO_2$ emissions appears to vary with the phases defined in the previous section, it does not depend on the ambient temperature in each phase, thereby indicating that the EGR and urea-SCR systems do not influence the amount of $CO_2$ emissions. The driving speeds in phases 1 and 2 were lower than that in phase 3 because the road tended to be crowded in phases 1 and 2, leading to the high emission of $CO_2$ caused by frequent acceleration. Moreover, in the beginning of the cold-start process, the engine is cooled over

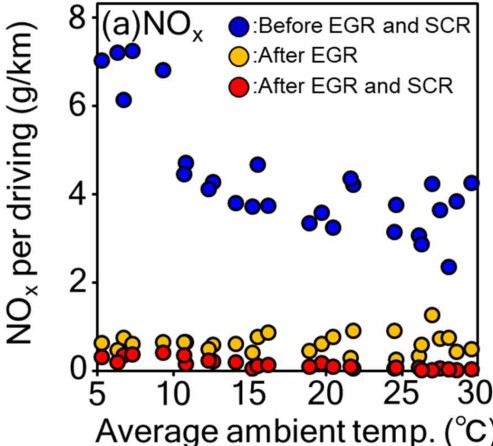

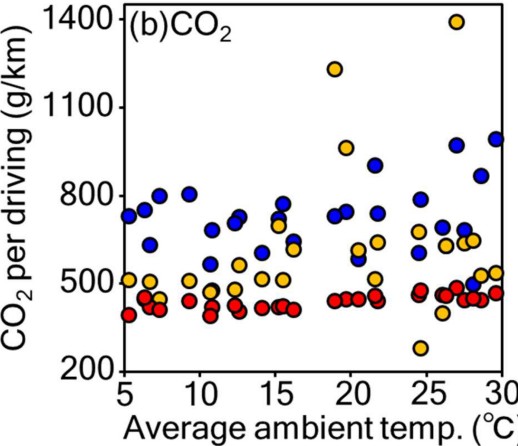

**Figure 2.** Relationships between ambient temperature and total amount of **(a)** $NO_x$ and **(b)** $CO_2$ emissions per driving distance (g/km). Each plot was obtained from morning and afternoon tests conducted in four seasons (the experiments were conducted for a period of 5 d in each season). Average ambient temperatures were determined using the average value of the temperatures recorded during each driving test.

several parking durations, and the combustion efficiency in this case is lower than that after the engine is warmed. Furthermore, in the cold-start process, the vehicle body is also cooled over several parking durations, and the friction of the rotatory parts of the vehicle is likely increased. Therefore, the difference in the $CO_2$ emissions in the three phases can be attributed to three cold-start features: difference in the vehicle speed in each phase, difference in the engine combustion efficiencies, and high rotation friction of the rotatory parts.

## 3.3 Comparison of $NO_x$ and $CO_2$ emissions determined using PEMS and chassis dynamometer

Figure S5 shows the vehicle speed and acceleration distribution determined using the PEMS measurement and WHVC mode from the chassis dynamometer measurement divided

into six torque ranges (0–100, 100–200, 200–300, 300–400, 400–500, and 500–600 N m) and three engine rotation ranges (500–1000, 1000–1500, 1500–2000 rpm). In the low to middle engine rotation ranges with middle to high torque ranges, the PEMS results correspond with a high acceleration in the low vehicle speed field, which was observed in all RDE test conditions. This indicates that real-world driving includes a sudden acceleration profile in the Japanese urban road which is not taken into account by the WHVC approach. We note that the acceleration distribution also depends on the driving feature, and the distribution changes for each driver (Ericsson, 2001). The drivers of the RDE and chassis dynamometer experiments were different in this study such that the results of the difference in the acceleration distribution can also be attributed to the difference in the driving feature. Moreover, the WHVC includes a high vehicle speed range which does not appear in the PEMS results. In this study, RDE was measured on an ordinary road in the metropolitan area of Tokyo. The vehicle speed on this road in the urban area was limited to 60 km/h, and thus data pertaining to speeds of more than 60 km/h were not collected because this study was focused on the $NO_x$ and $CO_2$ emission trends in urban areas. Obtaining the measurements for the high-speed range is a task for future work.

The $NO_x$ and $CO_2$ emissions per driving distance (g/km) in the four seasons, as obtained from the WHVC, are shown in Fig. 3. It can be noted that the emissions were almost the same in the four seasons, indicating that the vehicle condition in each season was nearly identical; therefore, the PEMS test results in the four seasons could be considered to be comparable. Although this study was focused on $NO_x$ and $CO_2$ emissions, the emissions of other gaseous species, including CO, THC, $CH_4$, and $N_2O$, were also measured, as summarized in Table S2.

## 3.4 Physical analysis of $NO_x$ and $CO_2$ emissions considering the vehicle speed and driving force

### 3.4.1 Force–speed–emission transient map

When evaluating the emission inventory from vehicles, it is necessary to formulate the emission factor of each pollutant, which can be used to evaluate the amount of emissions in arbitrary environmental conditions (in terms of the time, location, ambient temperature, etc.). Some researchers have attempted to formulate the amount of emissions from vehicles based on PEMS experiments (Bishop et al., 2019) and the vehicle-specific power method (Koupal et al., 2005). However, in the case of heavy-duty vehicles, the after-treatment system is complex, and the formulation of the pollutants' emissions involves high variability. To eliminate this variability from the estimation model, a transient emission table was constructed in this work. The formulation method was based on the assumption that the amount of emissions such as those of $NO_x$ and $CO_2$ from engine exhaust depends on the

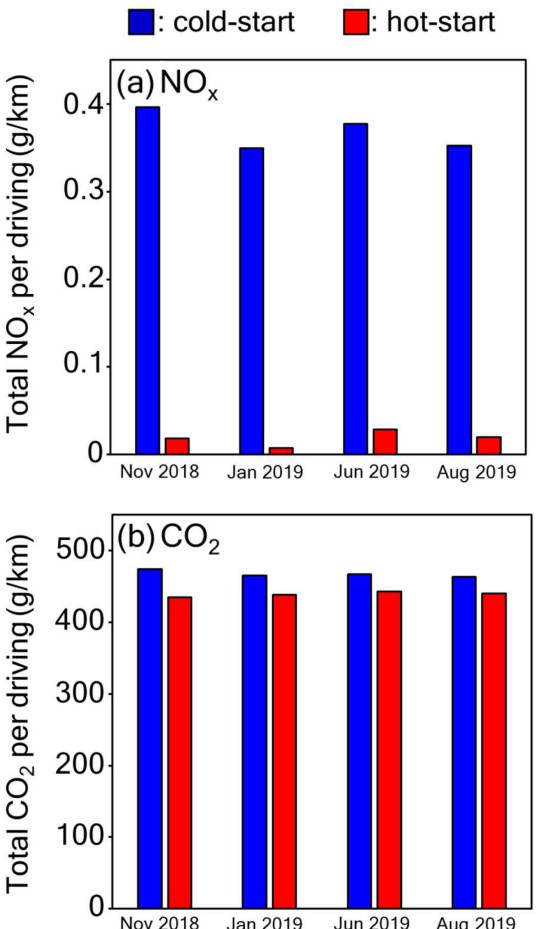

**Figure 3.** Experimental results of chassis dynamometer tests for world harmonized vehicle cycle cold and hot starts for **(a)** $NO_x$ and **(b)** $CO_2$ emissions.

driving force and vehicle speed at any given moment, consequently resulting in the transient $NO_x$ and $CO_2$ emissions from the tailpipe. Because this dependence cannot be formulated mathematically owing to the nonlinear relationship between the after-treatment tools of EGR and urea-SCR, an emission table containing the parameters of the driving force, vehicle speed, and amount of emissions was employed. The driving force was calculated using Eq. (4). The obtained transient table for the $NO_x$ and $CO_2$ emission generated by the tested vehicle and measured by PEMS in real-world driving is shown in Figs. 4 and 5, respectively. As shown in Fig. 4, the amount of $NO_x$ emissions is proportional to the driving force and vehicle speed and inversely proportional to the ambient temperature, as discussed in a previous section. The amount of $NO_x$ emissions decreases when the two $NO_x$ after-treatment systems, EGR and urea-SCR, start operating. According to Fig. 5, the $CO_2$ emissions increase in proportion to the driving force and vehicle speed; however, the amount of $CO_2$ emissions is not considerably influenced by the ambient temperature because the ambient tempera-

ture mainly affects the detoxification catalysis activity. After the EGR and urea-SCR start operating, the amount of $CO_2$ emissions in low-temperature conditions is lower than that in high-temperature conditions. When the measurement tests were conducted in the summer season, the air conditioner was switched on, while it was not used in the winter season. It was considered that the temperature dependence after the two $NO_x$ after-treatment systems started operating likely results from the use of the air conditioner. The use of the air conditioner leads to an additional load on the engine that may render engine operation at less optimum operational ranges, leading to an increase in emissions. Despite this, the trend of the air conditioner was not monitored in this study, and such discussions are one of the possibilities. The difference in the $CO_2$ emissions before and after the EGR and urea-SCR start operating is lower than that of the $NO_x$ emissions. In phase 1, the driving situation is similar to the cold start, and insufficient fuel combustion occurs. However, in phase 3, the driving situation is almost similar to that of a hot start, and the fuel combustion efficiency is maximized, resulting in lower $CO_2$ emissions. Using the transient tables mapped in Figs. 4 and 5, the $NO_x$ and $CO_2$ emissions from a test vehicle in an arbitrary driving pattern with an arbitrary temperature and road gradient can be predicted. Figure 6 shows the comparison between the $NO_x$ and $CO_2$ emissions determined experimentally via PEMS measurement and the values estimated from the transient emission tables mapped in Figs. 4 and 5. The transient table values agree with the experimental results with a correlation factor of 0.9. The results shown in Fig. 6 indicate that once the emission profiles of $NO_x$ and $CO_2$ have been obtained using a real-world driving method with detailed road information such as the ambient temperature and road gradient, the emissions in the arbitrary conditions can be predicted well. This aspect also applies to the profiles of other pollutants such as CO and THC; however, these pollutants are not the focus of this study. The dispersion of the plots might indicate the limitation associated with the application of the transient emission table in the estimation of $NO_x$ and $CO_2$, as well as other presumable vehicular exhaust emissions. The dispersion resulted from the fact that the transient emission table modeled in this study simulated real-time exhaust emissions based on driving force and vehicle speed. Real-time prediction is associated with several uncertainties that cannot be taken into account by the two parameters, driving force and vehicle speed, such as transient high acceleration, emission control system settings by the manufacturer, etc. Nevertheless, the predicted results shown in Fig. 6a and b include high linearity even for the real-time measurement results. The results also highlight the possibility of being applied in emission inventory evaluations in the future.

Moreover, in future work, the evaluation must be performed considering the high-speed range (more than 60 km/h) and more test vehicles. In this study, the RDE tests were conducted only in the urban area of Tokyo; however,

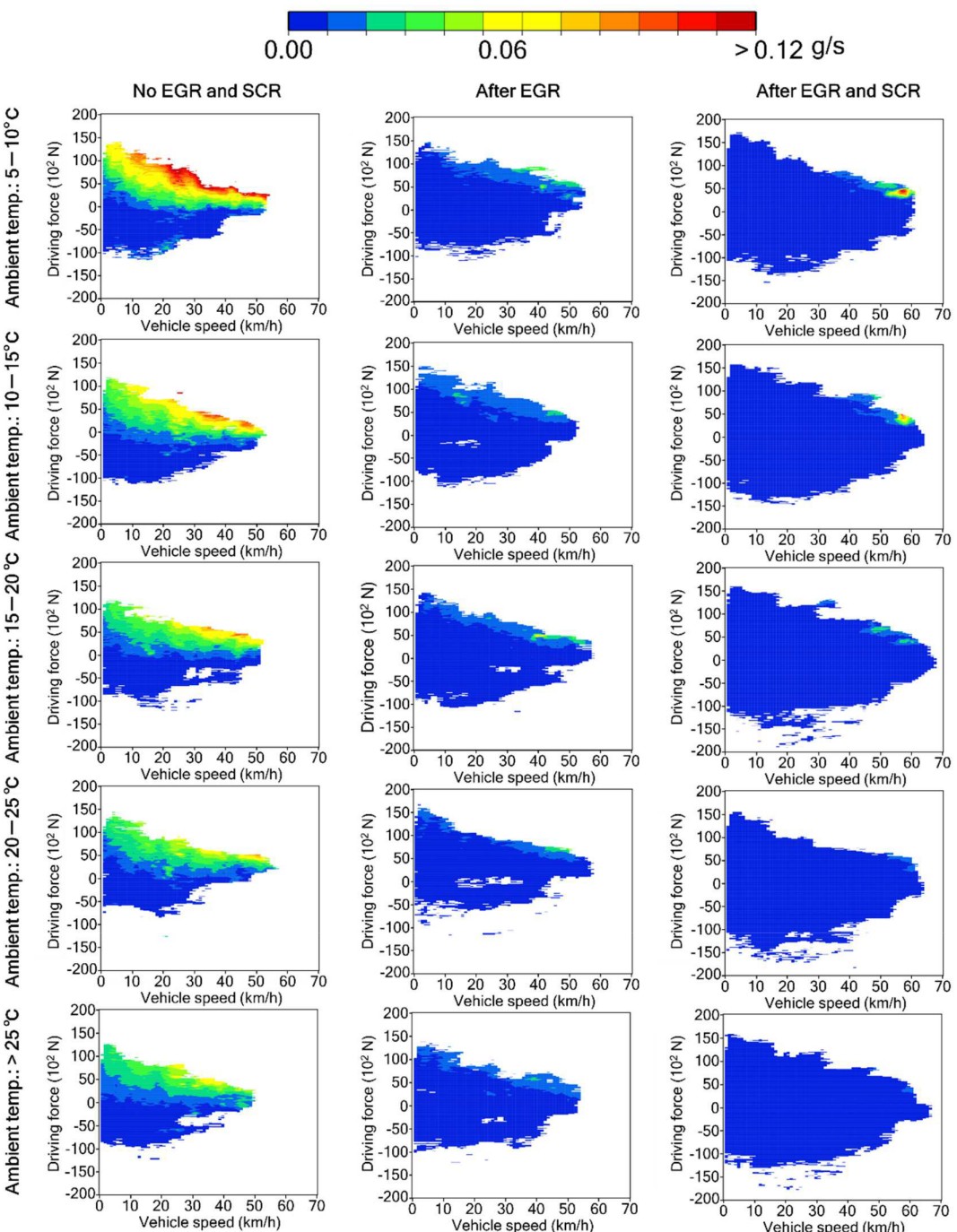

**Figure 4.** Transient emission map for $NO_x$ emissions evaluated using a real-world driving emission test conducted using PEMS, as well as the vehicle driving force and speed data.

the test must also be conducted on an express highway to determine the pollutants' emissions in the high-speed range to evaluate the complete Japanese emission inventory. Furthermore, the emissions of only one heavy-duty vehicle were measured in this work. In future testing, several other types of vehicles must be considered to obtain a statistically valid inventory.

### 3.4.2 Sensitivity analysis of driving force

The evaluated transient map indicated that the driving force directly influences the amount of $NO_x$ and $CO_2$ emissions. According to Eq. (4), the driving force includes three parameters that are related to the vehicle operation: acceleration, road gradient, and vehicle speed. To evaluate the parameters

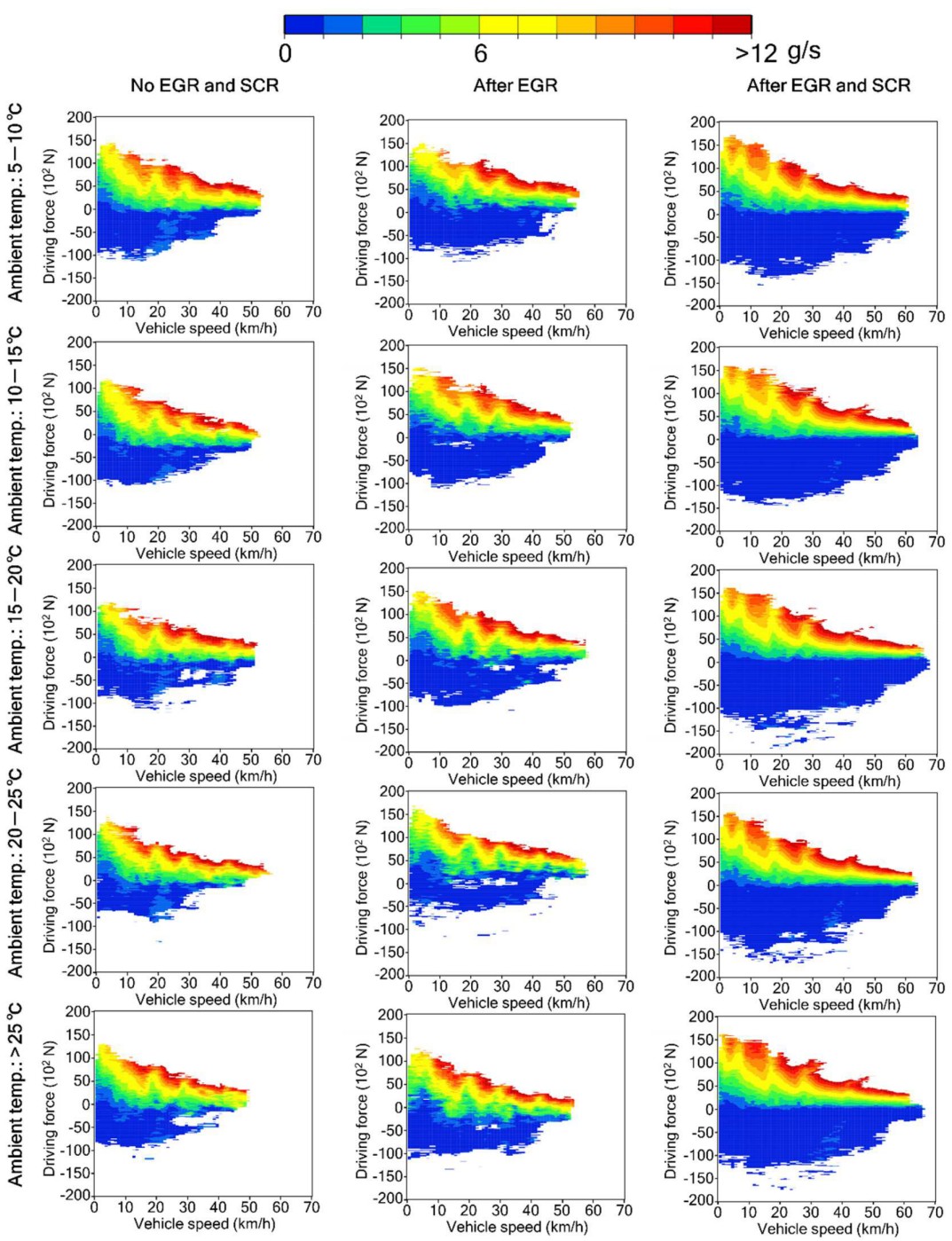

**Figure 5.** Transient emission map for CO$_2$ emissions evaluated using a real-world driving emission test conducted using PEMS, as well as the vehicle driving force and speed data.

that considerably influence the driving force in real-world driving, a sensitivity analysis was conducted based on the following sensitivity formulas derived using the partial differential of the three parameters.

$$\frac{\partial F}{\partial a} = m + \Delta m \tag{5}$$

$$\frac{\partial F}{\partial \theta} = mg\cos\theta \tag{6}$$

$$\frac{\partial F}{\partial v} = 2\mu_a A v \tag{7}$$

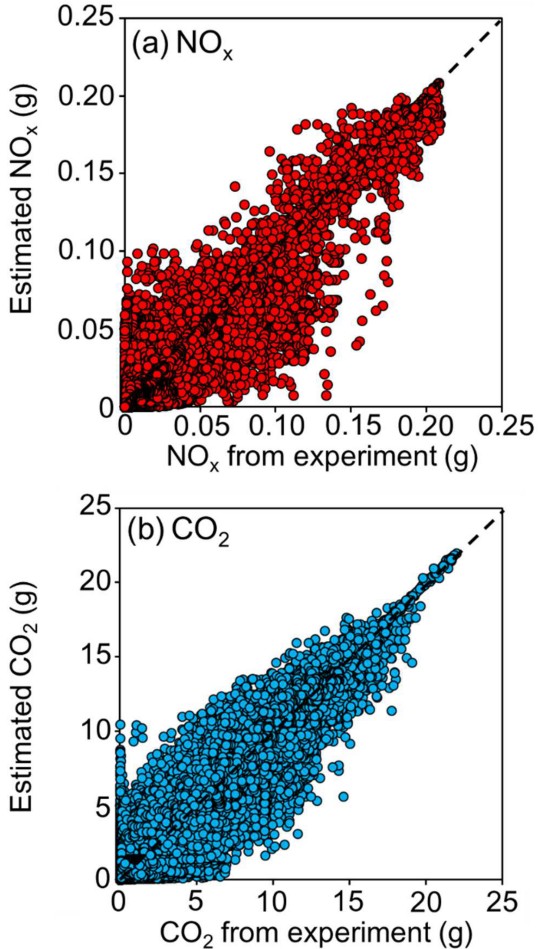

**Figure 6.** Correlation analysis of amount of **(a)** $NO_x$ and **(b)** $CO_2$ emissions measured and estimated using transient tables.

Equations (5), (6), and (7) indicate the sensitivities of the acceleration, road gradient, and vehicle speed, respectively. The definitions and values of the constant parameters related to the tested vehicle have been described in Sect. 2.3.2. According to Eq. (5), the sensitivity of the driving force to the acceleration is constant at 6468 N/(m s$^{-2}$). According to Eq. (6), the sensitivity of the driving force to the road gradient depends on the cosine of the road angle. The minimum and maximum absolute values of the road gradient in the driving course considered in this work were 0 % and 6.8 %, respectively, corresponding to a driving force sensitivity of 57 624 and 57 541 N, respectively. According to Eq. (7), the sensitivity of the driving force to the vehicle speed is dependent on the vehicle speed. The minimum and maximum vehicle speeds in the driving process in this work were 0 and approximately 70 km/h, corresponding to sensitivities of 0 and 0.80 N/(m s$^{-1}$), respectively. Based on the three sensitivity factors, the road gradient most notably influences the driving force and leads to an increase in the $NO_x$ and $CO_2$ emissions, as discussed in Sect. 3.4.1. In the laboratory test using the chassis dynamometer, the WHVC driving mode is currently applied worldwide, including Europe, the United States, and Japan, among others, mainly for research purposes (DieselNet Webpage, 2020 TS7). Although this mode includes the parameter of the road gradient, the road gradient strongly depends on the road characteristics, and therefore, it may be difficult to replicate the exhaust emission trend in a large area when using this approach. RDE monitoring is thus meaningful to evaluate the emissions in each specific area.

## 4   Conclusions

RDE experiments for a heavy-duty vehicle used in the Japanese market were conducted using PEMS, and the $NO_x$ and $CO_2$ emission trends were analyzed. The experimental results indicated that the amount of $NO_x$ emissions was higher in colder seasons owing to the effect of the two $NO_x$ after-treatment systems, EGR and urea-SCR. These systems starting operating earlier in warm seasons than in cold seasons, leading to a larger amount of emissions in colder seasons. The $CO_2$ emissions did not exhibit an apparent seasonal dependence; however, the amount of $CO_2$ emissions was relatively larger in colder seasons owing to the low engine combustion efficiency caused by the low ambient temperature. The speed and acceleration distributions pertaining to real-world driving tests using PEMS and WHVC driving mode from the chassis dynamometer experiments indicated that real-world driving in the urban area of Tokyo included a high acceleration in the low-speed range which is not reflected in the WHVC driving mode. The transient emission tables for $NO_x$ and $CO_2$ were constructed based on the experimental results and two parameters, i.e., the driving force and vehicle speed, which replicated well the PEMS experimental results. Consequently, these tables could be used to evaluate the $NO_x$ and $CO_2$ emission inventories. The results of the sensitivity analysis for the driving force suggested that the road gradient most notably influences the amount of $NO_x$ and $CO_2$ emissions, thereby demonstrating the importance of conducting RDE measurements which take into account the road characteristics in a specific area. The experiment and the detailed analysis were conducted only for one heavy-duty vehicle in the Japanese market. In the future, it is expected that further studies will be conducted to obtain the variability in real-world vehicular exhaust emissions; after the measurement results for a consistent number of vehicles have been obtained, the emissions inventory based on real-world measurements should be evaluated for use in policy making regarding air quality treatments.

*Data availability.* All the RDE data measured in this study are available upon request from the corresponding author.

*Supplement.* The supplement related to this article is available online at: https://doi.org/10.5194/amt-14-1-2021-supplement.

*Author contributions.* HH, MK, MY, and JH designed the research. HH, MK, KY, MO, CF, and MY performed the experiments. KK and TO analyzed the statistical data. HH, KK, and TO analyzed the experimental data. HH wrote the paper.

*Competing interests.* The authors declare that they have no conflict of interest.

*Acknowledgements.* This study was funded by the Bureau of Environment, Tokyo Metropolitan Government.

*Financial support.* This study was funded by the Bureau of Environment, Tokyo Metropolitan Government. APC CE5 was funded by the Tokyo Metropolitan Research Institute for Environmental Protection.

*Review statement.* This paper was edited by Anna Novelli and reviewed by Justin Bishop and one anonymous referee.

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

**Remarks from the language copy-editor**

CE1    Please verify the addition of "an".
CE2    Please verify the inclusion of "a".
CE3    Thank you for the clarification. Please verify the addition.
CE4    Thank you again for clarifying the term. Please verify its inclusion here in the caption.
CE5    Sorry, what does APC mean? Is it an abbreviated name?

**Remarks from the typesetter**

TS1    Please confirm.
TS2    Please note that it should be clear that it is a reference that can be found in the list, therefore the year is needed. Please advise for both instances.
TS3    Please check the other values as well. Should they also be "4.920" and "27.241"? Please note that in the manuscript, it says "5,193", "4,920" and "27,241". If you only want to change this value to "5.193", please give an explanation of why this needs to be changed. We have to ask the handling editor for approval. Thanks.
TS4    Please give an explanation of why this needs to be changed. We have to ask the handling editor for approval. Thanks.
TS5    Please give an explanation of why this needs to be changed. We have to ask the handling editor for approval. Thanks.
TS6    Please confirm. It should be clear that it is a reference that can be found in the list, therefore the year is needed.
TS7    Please confirm.
TS8    Please confirm.