# Peer review of "Real-world measurement and mechanical-analysis-based-verification of NOx and CO2 emissions from in-use heavy-duty vehicle"

_Atmospheric Measurement Techniques, 2020_

## Referee Comment (RC1) · Justin Bishop (Referee) · 22 Nov 2020

The authors have tested a heavy-duty vehicle in Japan on a dynamometer and using a PEMS to evaluate how driving force and season influences emissions of CO2 and NOx.

The literature review is not sufficient to show why this work is novel, especially as the authors measure one vehicle only.

The methodology is not transparent sufficiently to allow key outputs to be replicated, namely the transient emissions maps. There are issues with the figures in the SI which

need to be addressed, including the transient emissions maps and the correlation analysis.

Line 20: Define long-term and short-term Line 23: What year is the NASA reference? Line 23: Better to use 'climate change' instead of 'global warming' Line 25: This reference is 10 years old – can you find a newer source to support your point? Line 25: Is 'photochemical oxidant' a single species? It seems this would be a group of chemicals Line 41: I don't believe this to be the case – lab tests are set to standard conditions to allow repeatability over all tests. The narrow test conditions means the results may not align with what we see in more varied real-world conditions Line 44: What is the EPA reference year? Catalytic converters operate based on stoichiometric combustion in a spark ignition engine, and are (I believe) relatively independent of the ambient temperature Line 45: Might be better to say proportional to fuel burn, since exhaust treatment technology mitigates the effect of driving conditions on tailpipe emissions Line 70: What are the 2016 Japanese regulations? Line 81: How were the lab test conditions modified to reflect different seasons? Line 90: Why was EGR measured only in spring and summer? Line 91: I assume the route was the same across all days and seasons? Line 96: What is the justification for a 5 second smooth? Line 99: Worth explaining the central difference method and justifying its use here Line 120: Rolling resistance and aerodynamic drag should be derived from coast down tests – was this done in your 2012 work referenced? Line 121: You have switched from km/s2 to m/s2 units for acceleration. How is the 0.139 m/s2 threshold determined? Line 122: How was the test mass determined? Line 132: What is the justification for smoothing the altitudes? Altitudes are already smoothed to be a constant value within each mesh. Line 133: Similarly, what is the justification for the 7m smoothing to determine road slope? Line 150: I disagree - exhaust temperature varies more than coolant temperature Line 184: Should define the torque and speed ranges Line 204: There is no method to recreate the transient emissions table Line 206: Engine out emissions are related to driving force (and fuel used), but tailpipe NOx is decoupled from engine out emissions due to active management by the SCR Line 213: Earlier, you said ambient

temperature had an important role to play in emissions Line 218: Some evidence is needed for this. The air conditioner will manifest as some additional load, in the same way as a heater during the winter. Fig 1: There are two trips per day in each of the seasons – does this graph show the average of those two trips? Were all of these cold starts, with engine coolant temperature from the same starting point Fig 2: How is this graph determined? We don't know what the ambient temperatures were in each of the eight tests (two tests per day, four seasons) Fig 3: How many dyno tests were done? Were the ambient conditions of the PEMS test replicated here? Fig 4: Why is the area of the EGR + SCR plots (third column) larger than the No EGR + SCR and EGR only plots? They should all occupy the same area because vehicle speed and driving force doesn't change across the three columns Fig 5: As Fig 4 Fig 6: The R2 value for these graphs might be high, but there is large variation about the 1:1 line

---

## Author Comment (AC1) · 4 Dec 2020

The authors have tested a heavy-duty vehicle in Japan on a dynamometer and using a PEMS to evaluate how driving force and season influences emissions of CO2 and NOx. The literature review is not sufficient to show why this work is novel, especially as the authors measure one vehicle only. The methodology is not transparent sufficiently to allow key outputs to be replicated, namely the transient emissions maps. There are issues with the figures in the SI which need to be addressed, including the transient emissions maps and the correlation analysis.

The authors are grateful to the reviewer, Dr. Bishop, to taking a time and giving us

the important insights, suggestions and clarifications. We modified the manuscript carefully based on the opinions. The added and modified sentences were highlighted in yellow. To address the novelty and validity of the experiments and analysis (using only one heavy-duty vehicle), following sentences were added in the article. Lines 68-69: "To the best of our knowledge, this is the first study in which PEMS measurement results are applied in the development of an estimation model for vehicular exhaust." Lines 304-306: "The experiment as well as the detailed analysis were conducted only for one heavy-duty vehicle in the Japanese market. In future, it is expected that further studies would be conducted to obtain the variability of real-world vehicular exhaust emissions."

Line 20: Define long-term and short-term

According to the suggestion, definitions and examples were added as the following sentences. Line 20-22: "The air pollution caused by long-term air pollutants, which are chemically stable components such as CO2, and short-term air pollutants, which are reactive chemicals such as NOx, volatile organic compounds (VOCs), and photochemical oxidants, is a cause of significant concern in many countries."

Line 23: What year is the NASA reference? The webpage of NASA is updating continuously, so the concrete year was not defined in the citation. Meanwhile, we added the corresponding year of 1 degC increase from average temperature between 1951 and 1980 as "... in 2020" in Line 24.

Line 23: Better to use 'climate change' instead of 'global warming' Thank you for the suggestion. The terms were substituted by 'climate change'.

Line 25: This reference is 10 years old – can you find a newer source to support your point? According to the suggestion, we added IPCC report, "Summary for Policymakers", published online in 2018 as the citation which describes the potential increase of global temperature in the near future.

**AMTD**
Line 25: Is 'photochemical oxidant' a single species? It seems this would be a group of chemicals

Thank you for pointing out the lack of information. Photochemical oxidants are composed of several components including ozone, PAN etc. and approximately 90% of oxidants are ozone. The sentence was modified as following. Lines 27-28 "In addition, photochemical oxidants, mainly composed of ozone, are well-known short-term air pollutants, ..."

Line 41: I don't believe this to be the case – lab tests are set to standard conditions to allow repeatability over all tests. The narrow test conditions means the results may not align with what we see in more varied real-world conditions

The authors understood what the reviewer wanted to mention. There were several studies focusing on extreme temperature conditions such as exhaust measurements on the very low temperature conditions, but the laboratories which can conduct the kind of experiments were very limited mainly because of the construction cost of measurement setup. Meanwhile, we added the following sentences to further explain the pros aspect of PEMS experiment. Lines 43-44: "(recently however, temperature variable chassis dynamometers were adopted only in a limited number of laboratories (Clairotte et al., 2013; Ko et al., 2017)).". Lines 49-51: "Road temperature and gradient might vary from one season and location to another; therefore, real driving emission measurements are more suitable for a better understanding of the real-world vehicular exhaust verification."

Line 44: What is the EPA reference year? Catalytic converters operate based on stoichiometric combustion in a spark ignition engine, and are (I believe) relatively independent of the ambient Temperature

Thank you for pointing out the lack of information, the reference year 2010 was added in the citation. Also thank you for pointing out the mistake of the sentence. Ambient temperature affects to the amount of cold-start emission especially because of the Interactive comment

temperature dependence of engine coolant temperature and the urea injection time. These discussions are mentioned in section 3.1 in the main article. We corrected the sentence as following. Lines 44-47: "It has been noted that environmental temperature considerably influences the amount of exhaust emissions (detailed explanations regarding this observation are provided in section 3.1), leading to the release of a large amount of pollutants (including NOx) into the atmosphere in low ambient temperature conditions (U.S. Environmental Protection Agency, 2010)"

Line 45: Might be better to say proportional to fuel burn, since exhaust treatment technology mitigates the effect of driving conditions on tailpipe emissions Thank you for the suggestion. Based on the suggestion, the sentence was modified as following. Lines 48-49: "..., which is presumed to be proportional to fuel consumption (or CO2 emission) and other exhaust emissions..."

Line 70: What are the 2016 Japanese regulations? The detail of the regulation was added in Table S1 of supplementary information.

Line 81: How were the lab test conditions modified to reflect different seasons?

We did not reflect the seasonal dependencies by laboratory tests. The purpose of chassis dynamometer measurement includes two folds: First, understanding the differences of velocity and acceleration profiles between lab test and RDE, and 2nd, to verify as the same condition of the tested vehicle in each tested dates because the experiments were conducted in four different dates and we needed to assure the vehicle performance to be the same condition to compare the RDE results in the different tested dates. The results are mentioned in section 3.3 of main article. To mention those reasons, following sentence was added in the main article. Lines 75-78: "The purpose of the laboratory tests was to determine the difference between chassis dynamometer measurements and real-world driving emission measurement results. The tests were also performed to verify whether the conditions under which the vehicles performed were the same in the four different seasons that were investigated. This enabled a

AMTD
comparison of the seasonal dependencies of real-world measurements."

Line 90: Why was EGR measured only in spring and summer?

In the first two experiments, autumn (November 2018) and winter (January 2019), EGR ratio was not taken in account to be measured. After we analyzed the experimental results of those two seasons, we realized that it was important to measure EGR ratio and the measurement of EGR ration from OBD was conducted for remained two seasons.

Line 91: I assume the route was the same across all days and seasons?

Thank you for clarifying lack of information. Exactly the route was the same in all the experiments. This was added as following. Line 102: "The driving route was the same across all days and seasons."

Line 96: What is the justification for a 5 second smooth?

We added the 5 points smoothing flow as the following sentence. Lines 111-115: "In equation (1), vehicle speed was smoothed using the 5-point average of the speeds at neighbouring time steps. We determined the averaging number of vehicle speed as 5 points based on two concepts. First, the averaging number of vehicle speed should be minimized as much as possible to maintain the high resolution time step. Second, the dispersion should be sufficiently lower than the measured vehicle speed. It is also worth noting that the averaging number of vehicle speed may depend on the measurement tools, and the 5-point value was suitable in this study."

Line 99: Worth explaining the central difference method and justifying its use here

Central difference method is a mathematical discretization formula of the differential form which is described in equation (2) of main article. The vehicle speed measured by the experiments was not the continuous value, so the discretization was needed to obtain acceleration. We think that central difference method is just a mathematical form of discretization method and no justification can be added. Please reconsider our insights.

AMTD
Line 120: Rolling resistance and aerodynamic drag should be derived from coast down tests – was this done in your 2012 work referenced?

First, the authors are feeling sorry about the incorrect reference. The resistance parameters were not cited from our work in 2012. The parameters were set by using the same value of heavy-duty vehicle measured in our previous study (but there is no official citation), because the parameters were not derived from coast down tests in this study. The sentence was modified as following. Lines 135-137 "While the coast down test to determine  $\mu$ r and  $\mu$ a was not conducted in this study, based on our previous study,  $\mu$ r and  $\mu$ a were set at 0.0089 and 0.0027, respectively, given that we used the same type of heavy-duty vehicle that was tested in our previous study."

Line 121: You have switched from km/s2 to m/s2 units for acceleration. How is the 0.139 m/s2 threshold determined? Line 122: How was the test mass determined?

The unit of vehicle acceleration defined in section 2.3.1 is described as km/s2 because the monitored vehicle speed was in the unit of km/s, on the purpose of unifying the unit. On the other hand, m/s2 is used in section 2.3.3 because acceleration parameter is used to calculate driving force, the unit is defined in N(=kg m/s2). 0.139 m/s2 corresponds to 0.5 km/(h s). The unit km/(h s) can be considered as the differential of vehicle speed (km/h) by time (s), dv/dt. In this study, we defined 0.5 km/(h s) or less than this value as almost no acceleration because this value is considered to be low enough to be considered as zero. The boundary threshold, 0.5 km/(h s) was used in the previous study (Yoshizumi et al., 1980), and the reference was also added in our manuscript. Yoshizumi, K et al. Automotive Exhaust Emission in an Urban Area. SEA. Tech. Pap. Ser. 1980, 800326, p17.

Line 132: What is the justification for smoothing the altitudes? Altitudes are already smoothed to be a constant value within each mesh.

We added the following explanation about how the smoothing was conducted. Lines 150-152: "...This smoothed value in the vicinity of  $\sim$ 5-m meshes was determined by
varying the averaging mesh number (e.g., from 1, 2 ...), while carefully checking to ensure that the noise remained negligible relative to the height change."

Line 133: Similarly, what is the justification for the 7m smoothing to determine road slope? We added the following sentence as justification of smoothing value. Lines 154-155: "This smoothed value, in the vicinity of  $\sim$ 7-m meshes, was determined using the same method that was applied to altitude data, as described above."

Line 150: I disagree - exhaust temperature varies more than coolant temperature

Thank you for pointing out the irrelevant description. The sentence was modified as following. Lines 171-173: "Figure S1 suggests that in the cold-start situation, the engine coolant temperature was proportional to the exhaust gas temperature. Further, in general, engine coolant temperature is usually used to control the EGR system."

Lines 206-207: Should define the torque and speed ranges

According to the suggestion, the sentence was modified. Lines 195-196: "...divided into six torque ranges (0–100, 100–200, 200–300, 300–400, 400–500, and 500–600 N m) and three engine rotation ranges (500–1000, 1000–1500, 1500–2000 rpm)."

Line 204: There is no method to recreate the transient emissions table

The authors were convinced about the opinion above. All the raw data of temperatures, NOx and CO2 emissions, road gradients etc. have been sorted to be published. Because of the large size of the whole data, we are now preparing the database with DOI. Please wait for few weeks.

Line 206: Engine out emissions are related to driving force (and fuel used), but tailpipe NOx is decoupled from engine out emissions due to active management by the SCR

Thank you for the clarification. To explain the precise meaning and background of transient emission table, the related sentences were improved as following. Lines 227-232: "The formulation method was based on the assumption that the amount of emis-
sions such as those of NOx and CO2 from engine exhaust depend on the driving force and vehicle speed at any given moment, consequently resulting in the transient NOx and CO2 emissions from the tailpipe. Because this dependence cannot be formulated mathematically owing to the non-linear relationship between the detoxification tools of EGR and urea-SCR, an emission table containing the parameters of the driving force, vehicle speed, and amount of emissions was employed."

Line 213: Earlier, you said ambient temperature had an important role to play in emissions

At this sentence, the authors intended to mention that temperature is not critical to "CO2" emission, and the term "CO2" was added in the revised manuscript.

Line 218: Some evidence is needed for this. The air conditioner will manifest as some additional load, in the same way as a heater during the winter.

Unfortunately, the behavior of air conditioner was not monitored in this study. For this reason, we added additional information as below. Thank you for your reconsideration. Line 241: "...while it was not used in the winter season, ..." Lines 243-244: "Despite this, the trend of the air conditioner was not monitored in this study, and such discussions are one of the possibilities."

Fig 1: There are two trips per day in each of the seasons – does this graph show the average of those two trips? Were all of these cold starts, with engine coolant temperature from the same starting point

Thank you for pointing out the lack of information. The time trends shown in Fig.1 are AM test results: this information was added in the caption of Fig.1. The start point of coolant temperatures were different in each season, this is also one of the reasons why the seasonal dependency happened. We added this discussion as following. Lines 176-177: "...because the initial coolant temperature is higher in the hotter seasons."

Fig 2: How is this graph determined? We don't know what the ambient temperatures

**AMTD**
were in each of the eight tests (two tests per day, four seasons)

Thank you for pointing out the lack of information. The detail of temperatures and plots were added in the caption of Fig.2 as following. "Each plot was obtained from AM and PM tests conducted in four seasons (the experiments were conducted for a period of 5 days in each season). Average ambient temperatures were determined using the average value of the temperatures recorded during each driving test."

Fig 3: How many dyno tests were done? Were the ambient conditions of the PEMS test replicated here?

One chassis dynamometer test was conducted before the PEMS experiment in four seasons, so totally four dyno tests were conducted. The ambient conditions were almost the same between all the tested environments. We added this information in Lines 90-91. Lines 92-93: "n all the laboratory measurements, room temperature was set to be approximately 25  $^{\circ}$ C."

Fig 4: Why is the area of the EGR + SCR plots (third column) larger than the No EGR + SCR and EGR only plots? They should all occupy the same area because vehicle speed and driving force doesn't change across the three columns

Driving force and vehicle speed are different between three phases. No EGR+SCR results were obtained from first 10 min after the driving started. On the other hand, EGR+SCR results were obtained from 30 min after the driving started. The driving root after 30 min included high speed and road gradient environments, and therefore, EGR+SCR results holds wide range of driving and speed compared with other phases.

Fig 5: As Fig 4 Fig 6: The R2 value for these graphs might be high, but there is large variation about the 1:1 line

The authors added the following sentence to refer the dispersion of predicted amount of exhaust emissions as for the reviewer's opinion. Lines 254-261: "The dispersion of the plots might indicate the limitation associated with the application of the transient

AMTD
emission table in the estimation of NOx and CO2, as well as other presumable vehicular exhaust emissions. The dispersion resulted from the fact that the transient emission table modelled in this study simulated real-time exhaust emissions based on driving force and vehicle speed. Real-time prediction is associated with several uncertainties that cannot be taken into account by the two parameters, driving force and vehicle speed, such as transient high acceleration, emission control systems settings by the manufacturer, etc. Nevertheless, the predicted results shown in Figure 6(a) and (b) include high linearity even for the real-time measurement results. The results also highlight the possibility of being applied in emission inventory evaluation in future."

Please also note the supplement to this comment: https://amt.copernicus.org/preprints/amt-2020-286/amt-2020-286-AC1supplement.zip

---

## Referee Comment (RC2) · Nikiforos Zacharof (Referee) · 6 Dec 2020

**1 General comments**

**1.1 Introduction**

The authors should pay more attention to the introduction and make clear what is the purpose of the paper. There are several statements about the representativeness of the laboratory testing compared to the on-road measurements, but it is not clear why this is discussed, while both approaches have pros and cons. Without proper listing

of them cannot enable a comparison between the two approaches. In this section, it is needed to clarify better the goals of the paper and why the authors conducted this research.

**1.2 Methodology**

Please describe the methodological approach (testing, analyses, etc) by offering an overview before moving into describing the testing process. With this structure, the reader is confused as the text moves directly into explaining the testing, but there has been no frame to put it into. Also, a clear description of the post-processing and the analyses that are performed to reach the goal of the paper is missing. Please provide an overview of the approach, describe each step of the approach and indicate all the analyses that you've performed. Additionally, please avoid presenting equations that a reader would already be familiar to, e.g. formula of acceleration.

1.3 Results and discussion

The authors need to improve the presentation of the results in terms of text structure and language. Several times it has been unclear what they wanted to say, while some other times the text was repetitive. Please present all the analyses that help you answer those questions that you have set. At this point, it seems only a simple presentation of the measurement results. This is useful to enhance the literature in the topic, but it does not add up significantly, while there is potential for this. Please consider adding a more detailed sensitivity analysis, an energy analysis (energy share on wheel, auxiliaries, etc) and an improved comparison between the WHVC and on-road conditions. These are just recommendations to improve the paper, based on what is already included. Consider expanding the analyses as you see fit. Interactive comment

**1.4 Conclusions**

The conclusion sector is unclear to what outcome the analysis reached. This section should avoid repeating what has already been presented in the discussion and actually reach to some conclusions and evaluate whether the paper's goals have been achieved or not, what new questions have arisen and what could be done next. The sole outcome is that the road gradient is the most important factor affecting NOx and CO2 emissions, which is hardly new. The authors conducted an experimental campaign that should gain visibility, but it needs more analyses, reach some solid conclusions that could actually contribute in the field.

**2 Specific comments**

Line 20 Please provide references and/or examples of major countries/regions that are concerned with pollutants.

Line 23 Please provide reference to the NASA report or database.

Line 25 The Saito 2010 reference is already 10 years old. Please add also a projection from newer studies if possible.

Line 25 - 26 "...photochemical oxidant is a well known short-term air pollutant..." It is not clear which photochemical oxidant the authors refer to and also there's lack of any reference to back this statement. Please elaborate on the whole sentence.

Line 30-31 "To address the problems of global warming and photochemical pollutants, it is necessary to mitigate air pollution." This statement is not quite accurate. Pollutants cause health problems mainly and also sometimes have a global warming potential. The major factor for global warming from the automotive sector are CO2 emissions. CO2 is not a pollutant and it doesn't cause health problems by inhaling it. Please
elaborate.

Line 40-41 "In general, the laboratory temperature is set at approximately 25 âDČ, and it cannot 40 be easily changed via the normal laboratory system." This is not quite true, the temperature in the laboratory and more specifically in the vehicle test cell can be adjusted to a range of different temperatures. In some cases, however, it could require additional investment in infrastructure such as to achieve a temperature of -7 oC that are required in some countries. The 25oC temperature is mandated by the testing protocol and the regulation that has adopted this protocol and not necessarily from the technical capabilities of the laboratory. Please elaborate the sentence.

Line 43 "...a specific activation temperature that cannot be attained in cold seasons...", this is quite a bold statement and not entirely true. The activation temperature will be reached at some point, but under cold conditions, it could take longer. In the case of small trips and low temperatures, then it is possible that the activation temperature is not reached. Please elaborate.

Line 44 – 46 "Moreover, the road gradient also influences the amountof exhaust emissions because it directly affects the driving force, which is presumed to be proportional to CO2 and other exhaust emissions." This statement is not correct. The gradient increases the required engine load and it indeed increases CO2 emissions, but this is not done necessarily linearly. The road gradient force formula is defined as  $m^*g^*sin(a)$  (m = vehicle mass, g = acceleration of gravity, a = road grade). The engine needs to operate at a higher load to compensate for this force, but the operation point depends also on the transmission ratio. In any case, the road grade increases fuel consumption and therefore CO2 emissions – not necessarily linearly. In the case of pollutants, they could increase but this depends on the engine operation point that affects what kind of pollutants are produced in the engine. Whether these pollutants make it to the exhaust relies heavily on the operation of the aftertreatment systems.

Line 46 "Consequently..." This statement does not exactly explains why the govern-
ments are making PEMS tests and it seems like a leap of thought. Also no references are provided to back this statement. The following statement in line 48 "Gallus et al. (2017)..." refers to an experimental campaign and not to the adoption of PEMS as an official protocol to define emissions. The following references also refer to research campaigns and not protocols. Please re-phrase and elaborate.

Line 55 "Nevertheless, the conduction of road measurement experiments using PEMS is a relatively new domain, and only a few studies have been performed to assess the analytical data..." This is not quite true, please check the literature on this issue an elaborate. PEMS is common in testing, especially on light-duty vehicles, where in the European Union at least is part of the vehicle type approval procedure for pollutants requires Real Driving Emission (RDE) testing (Regulation (EU) 2017/1151). The cold start is important for light-duty vehicles as they are often doing short trips and the aftertreatment systems do not reach always optimal operation temperature. However, for heavy-duty vehicles the cold start effect is limited as the vehicles operate for long time (e.g. 8 hours or longer for a typical city bus operation) and they are compensated by the long operation time.

Line 60 The term "classical mechanics" is a bit redundant in this context. Please consider to elaborate.

Line 62 Please take into consideration the instrument accuracy. PEMS can be utilized everywhere on-road, but could face accuracy issues, while these problems are minimized in the laboratory where the methodology and the instruments (e.g. bag result analysis) could be more accurate.

Line 98 The Eq. 1 seems redundant, but it could be retained. However, what the authors are describing here is a rolling average with a step of 5 observations. Please clarify.

Line 101 The Eq. 2 seems redundant as it presents the calculation of acceleration based on speed, which must be a well-known topic for the reader. Retain it if you
consider it useful for your narration, otherwise please remove and elaborate.

Line 103 The authors in this section (2.3.2) describe the synchronization of the data. For every data that are retrieved from the vehicle there's a time lag between the phenomenon that occurs and the measurement time. Combustion process and exhaust emissions are directly correlated as the latter are products of the former process but the data are retrieved with a time lag as gases need to travel from the combustion chamber to the exhaust where they are measured. Data timestamps from the combustion chamber and the exhaust would correspond to different events and they need to be synchronized. This applies to all the sensors but not all the events can be correlated in this way. This section seems that it is not needed as this process is standard for the post-processing. However, if you retain it please elaborate this section and consider merging it with another one where you describe the data post-processing.

Line 120 The  $\mu$ r and  $\mu\alpha$  are not defined. Please define that are the rolling resistance and air drag coefficients.

Line 122 You mentioned the vehicle weight was set at 5880. Have you measured the vehicle on a balance or did you derive this value from your calculations?

Line 142 The "engine room" could mean the whole compartment where the engine is placed. Please consider replacing it with the more appropriate term "combustion chamber".

Line 186 "which is not taken into account by the WHVC approach.". The WHVC development was based on real-world data in order to produce a representative situation of real-world conditions. The claim that the testing conditions are not represented in the WHVC is useful as observation, but it needs to back it with enough data. First, it should be quantified how many times are encountered the testing conditions that were outside the WHVC approach. The following questions must be answered. Has it been on every test, has it been affected by the driving style or has it been due to the requirements of your experiment? Second, it is needed to compare to real-world route conditions to the AMTD
overall WHCV approach and quantify the effect and its significance.

Line 199-207 This is methodology, please move to the respective part.

Line 217 - 218 It should be clarified that the use of air conditioning poses an additional load to the engine that could make the engine operate in less optimum operation ranges under some conditions. In this way it could lead to an increase in pollutants.

Line 245 - 250 It is stated several times the that "a parameter depends on the same parameter" such as "the road gradient depends on the cosine of the road gradient", which is quite obvious. Please elaborate the whole text and remove such statements.

Line 253 – 254 "the WHVC driving mode is currently applied worldwide." I am not sure whether this is true and it is quite vague. Please mention major countries/regions that use this protocol and for what reasons. In Europe for instance, the heavy-duty vehicle type approval procedure is performed through a simulatory approach that utilizes other driving cycles.

**3 Technical corrections**

Line 28 "(O'Neill et al., 2004; Chappelka and Samuelson, 1998; Wang et al., 2017)" please re-arrange in chronological order.

Line 44 Please provide an accurate reference for the EPA.

Line 54 "(Kousoulidou et al, 2013; Kwon et al, 2017; Liu et al, 2009; Luján et al, 2018; O'Driscoll et al, 2016)" please re-arrange in chronological order.

Line 70 "current Japanese regulation set in 2016." Please cite the exact law. Line 76 "...in our previous work..." Please avoid using possessive pronouns and especially in the first person. Consider removing them entirely and retain the reference to your work. If you want to retain the connection with your work for any reason, please consider
using the third person e.g. "in the authors' previous work".

Line 259 "in the Japanese market were conducted", I think you mean a vehicle that is available in the Japanese market, but it unclear. Please correct.

---

## Author Comment (AC2) · 29 Dec 2020

The authors have tested a heavy-duty vehicle in Japan on a dynamometer and using a PEMS to evaluate how driving force and season influences emissions of CO2 and NOx. The literature review is not sufficient to show why this work is novel, especially as the authors measure one vehicle only. The methodology is not transparent sufficiently to allow key outputs to be replicated, namely the transient emissions maps. There are issues with the figures in the SI which need to be addressed, including the transient emissions maps and the correlation analysis.

The authors are grateful to the reviewer, Dr. Bishop, to taking a time and giving us the

important insights, suggestions and clarifications. We modified the manuscript carefully based on the opinions. The added and modified sentences based on reviewer 1's opinion were highlighted in yellow. To address the novelty and validity of the experiments and analysis (using only one heavy-duty vehicle), following sentences were added in the article.

Lines 80-81: "To the best of our knowledge, this is the first study in which PEMS measurement results are applied in the development of an estimation model for vehicular exhaust."

Lines 327-331: "The experiment as well as the detailed analysis were conducted only for one heavy-duty vehicle in the Japanese market. In future, it is expected that further studies would be conducted to obtain the variability of real-world vehicular exhaust emissions; after the measurements results for the consistent number of vehicles have been obtained, the emissions inventory based on real-world measurements should be evaluated for use in policy making regarding air quality treatments."

Line 20: Define long-term and short-term

According to the suggestion, definitions and examples were added as the following sentences.

Line 20-23: "The air pollution caused by long-term air pollutants, which are chemically stable components such as CO2, and short-term air pollutants, which are reactive chemicals such as NOx, volatile organic compounds (VOCs), and photochemical oxidants, is a cause of significant concern in many countries, including the United States, the European Union, China, India, and Japan (Akimoto et al., 2015; Costa et al., 2014; Ravindra et al., 2016; Sullivan et al., 2018; Yang and Wang, 2017)."

Line 23: What year is the NASA reference?

The webpage of NASA is updating continuously, so the concrete year was not defined in the citation. Meanwhile, we added the corresponding year of 1 degC increase from

average temperature between 1951 and 1980 as "... in 2020" in Line 25.

Line 23: Better to use 'climate change' instead of 'global warming'

Thank you for the suggestion. The terms were substituted by 'climate change'.

Line 25: This reference is 10 years old – can you find a newer source to support your point? According to the suggestion, we added IPCC report, "Summary for Policymakers", published online in 2018 as the citation which describes the potential increase of global temperature in the near future.

Line 25: Is 'photochemical oxidant' a single species? It seems this would be a group of chemicals

Thank you for pointing out the lack of information. Photochemical oxidants are composed of several components including ozone, PAN etc. and approximately 90% of oxidants are ozone. The sentence was modified as following.

Lines 29-32 "In addition, photochemical oxidants, mainly composed of ozone, are well-known short-term air pollutants, generated by the photochemical reaction of NOx and volatile organic compounds (VOCs) (Sillman, 1999). The concentration of photochemical oxidants in the atmosphere is a significant concern for humans, animals, and crops in many countries (Chappelka and Samuelson, 1998; O'Neill et al., 2004; Wang et al., 2017)."

Line 41: I don't believe this to be the case – lab tests are set to standard conditions to allow repeatability over all tests. The narrow test conditions means the results may not align with what we see in more varied real-world conditions

The authors understood what the reviewer wanted to mention. There were several studies focusing on extreme temperature conditions such as exhaust measurements on the very low temperature conditions, but the laboratories which can conduct the kind of experiments were very limited mainly because of the construction cost of measurement setup. Meanwhile, we added the following sentences to further explain the pros aspect

of PEMS experiment.

Lines 44-45: "(recently however, temperature variable chassis dynamometers were adopted only in a limited number of laboratories (Clairotte et al., 2013; Ko et al., 2017))."

Lines 52-54: "Road temperature and gradient might vary from one season and location to another; therefore, real driving emission measurements are more suitable for a better understanding of the real-world driving emission (RDE) verification."

Line 44: What is the EPA reference year? Catalytic converters operate based on stoichiometric combustion in a spark ignition engine, and are (I believe) relatively independent of the ambient Temperature

Thank you for pointing out the lack of information, the reference year 2010 was added in the citation. Also thank you for pointing out the mistake of the sentence. Ambient temperature affects to the amount of cold-start emission especially because of the temperature dependence of engine coolant temperature and the urea injection time. These discussions are mentioned in section 3.1 in the main article. We corrected the sentence as following.

Lines 47-50: "It has been noted that environmental temperature considerably influences the amount of exhaust emissions (detailed explanations regarding this observation are provided in section 3.1), leading to the release of a large amount of pollutants (including NOx) into the atmosphere in low ambient temperature conditions (the MOVES2010 Report by the U.S. Environmental Protection Agency, 2010)."

Line 45: Might be better to say proportional to fuel burn, since exhaust treatment technology mitigates the effect of driving conditions on tailpipe emissions

Thank you for the suggestion. Based on the suggestion, the sentence was modified as following.

Lines 51-52: "..., which is presumed to have a negative effect on fuel consumption (or

CO2 emission) and other exhaust emissions."

Line 70: What are the 2016 Japanese regulations?

The detail of the regulation was added in Table S1 of supplementary information.

Line 81: How were the lab test conditions modified to reflect different seasons?

We did not reflect the seasonal dependencies by laboratory tests. The purpose of chassis dynamometer measurement includes two folds: First, understanding the differences of velocity and acceleration profiles between lab test and RDE, and 2nd, to verify as the same condition of the tested vehicle in each tested dates because the experiments were conducted in four different dates and we needed to assure the vehicle performance to be the same condition to compare the RDE results in the different tested dates. The results are mentioned in section 3.3 of main article. To mention those reasons, following sentence was added in the main article.

Lines 75-79: "The purpose of this study is two-fold. First, chassis-dynamometer-based and RDE measurements using PEMS were conducted on a heavy-duty Japanese vehicle to determine the importance of RDE specific factors, including the ambient temperature and road gradient, among others. Second, the obtained experimental results were analysed based on two parameters, i.e., the driving force and vehicle speed, to develop an analytical method to evaluate the amount of CO2 and NOx emissions from the vehicle in an arbitrary driving condition."

Line 90: Why was EGR measured only in spring and summer?

In the first two experiments, autumn (November 2018) and winter (January 2019), EGR ratio was not taken in account to be measured. After we analyzed the experimental results of those two seasons, we realized that it was important to measure EGR ratio and the measurement of EGR ration from OBD was conducted for remained two seasons.

Line 91: I assume the route was the same across all days and seasons?

Thank you for clarifying lack of information. Exactly the route was the same in all the experiments. This was added as following. Line 119: "The driving route was the same across all days and seasons."

Line 96: What is the justification for a 5 second smooth?

We added the 5 points smoothing flow as the following sentence.

Lines 111-115: "In equation (1), vehicle speed was smoothed using the 5-point average of the speeds at neighbouring time steps. We determined the averaging number of vehicle speed as 5 points based on two concepts. First, the averaging number of vehicle speed should be minimized as much as possible to maintain the high resolution time step. Second, the dispersion should be sufficiently lower than the measured vehicle speed. It is also worth noting that the averaging number of vehicle speed may depend on the measurement tools, and the 5-point value was suitable in this study."

Line 99: Worth explaining the central difference method and justifying its use here

Central difference method is a mathematical discretization formula of the differential form which is described in equation (2) of main article. The vehicle speed measured by the experiments was not the continuous value, so the discretization was needed to obtain acceleration. We think that central difference method is just a mathematical form of discretization method and no justification can be added. Please reconsider our insights.

Line 120: Rolling resistance and aerodynamic drag should be derived from coast down tests – was this done in your 2012 work referenced?

First, the authors are feeling sorry about the incorrect reference. The resistance parameters were not cited from our work in 2012. The parameters were set by using the same value of heavy-duty vehicle measured in our previous study (but there is no official citation), because the parameters were not derived from coast down tests in this study. The sentence was modified as following.

Lines 152-154 "While the coast down test to determine $\mu$r and $\mu$a was not conducted in this study, based on our previous study, $\mu$r and $\mu$a were set at 0.0089 and 0.0027, respectively, given that we used the same type of heavy-duty vehicle that was tested in our previous study."

Line 121: You have switched from km/s2 to m/s2 units for acceleration. How is the 0.139 m/s2 threshold determined? Line 122: How was the test mass determined?

The unit of vehicle acceleration defined in section 2.3.1 is described as km/s2 because the monitored vehicle speed was in the unit of km/s, on the purpose of unifying the unit. On the other hand, m/s2 is used in section 2.3.3 because acceleration parameter is used to calculate driving force, the unit is defined in N(=kg m/s2). 0.139 m/s2 corresponds to 0.5 km/(h s). The unit km/(h s) can be considered as the differential of vehicle speed (km/h) by time (s), dv/dt. In this study, we defined 0.5 km/(h s) or less than this value as almost no acceleration because this value is considered to be low enough to be considered as zero. The boundary threshold, 0.5 km/(h s) was used in the previous study (Yoshizumi et al., 1980), and the reference was also added in our manuscript.

Yoshizumi, K et al. Automotive Exhaust Emission in an Urban Area. SEA. Tech. Pap. Ser. 1980, 800326, p17.

Line 132: What is the justification for smoothing the altitudes? Altitudes are already smoothed to be a constant value within each mesh.

We added the following explanation about how the smoothing was conducted.

Lines 168-170: "...This smoothed value in the vicinity of $\sim$5-m meshes was determined by varying the averaging mesh number (e.g., from 1, 2 ...), while carefully checking to ensure that the noise remained negligible relative to the height change."

Line 133: Similarly, what is the justification for the 7m smoothing to determine road slope?

We added the following sentence as justification of smoothing value.

Lines 172-173: "This smoothed value, in the vicinity of $\sim$7-m meshes, was determined using the same method that was applied to altitude data, as described above."

Line 150: I disagree - exhaust temperature varies more than coolant temperature

Thank you for pointing out the irrelevant description. The sentence was modified as following.

Lines 189-191: "Figure S1 suggests that in the cold-start situation, the engine coolant temperature was proportional to the exhaust gas temperature. Further, in general, engine coolant temperature is usually used to control the EGR system."

Lines 206-207: Should define the torque and speed ranges

According to the suggestion, the sentence was modified.

Lines 224-225: "...divided into six torque ranges (0–100, 100–200, 200–300, 300–400, 400–500, and 500–600 N m) and three engine rotation ranges (500–1000, 1000–1500, 1500–2000 rpm)."

Line 204: There is no method to recreate the transient emissions table

The authors were convinced about the opinion above. All the raw data of temperatures, NOx and CO2 emissions, road gradients etc. have been sorted to be available to the readers. The data is vested in the Tokyo Metropolitan Government and we do not have the right to open the data to the public by online repository (after the discussion with the government). Therefore, the data could be available by the request to the corresponding author. The statement of data availability was added as the following sentence.

Line 333: "Data availability. All the RDE data measured in this study were available by the request for the corresponding author."

Line 206: Engine out emissions are related to driving force (and fuel used), but tailpipe NOx is decoupled from engine out emissions due to active management by the SCR

Thank you for the clarification. To explain the precise meaning and background of transient emission table, the related sentences were improved as following.

Lines 249-253: "The formulation method was based on the assumption that the amount of emissions such as those of NOx and CO2 from engine exhaust depend on the driving force and vehicle speed at any given moment, consequently resulting in the transient NOx and CO2 emissions from the tailpipe. Because this dependence cannot be formulated mathematically owing to the non-linear relationship between the detoxification tools of EGR and urea-SCR, an emission table containing the parameters of the driving force, vehicle speed, and amount of emissions was employed."

Line 213: Earlier, you said ambient temperature had an important role to play in emissions

At this sentence, the authors intended to mention that temperature is not critical to "CO2" emission, and the term "CO2" was added in the revised manuscript.

Line 218: Some evidence is needed for this. The air conditioner will manifest as some additional load, in the same way as a heater during the winter.

Unfortunately, the behavior of air conditioner was not monitored in this study. For this reason, we added additional information as below. Thank you for your reconsideration.

Line 262: "...while it was not used in the winter season, ..."

Lines 265-266: "Despite this, the trend of the air conditioner was not monitored in this study, and such discussions are one of the possibilities."

Fig 1: There are two trips per day in each of the seasons – does this graph show the average of those two trips? Were all of these cold starts, with engine coolant temperature from the same starting point

Thank you for pointing out the lack of information. The time trends shown in Fig.1 are AM test results: this information was added in the caption of Fig.1. The start point of coolant temperatures were different in each season, this is also one of the reasons why the seasonal dependency happened. We added this discussion as following.

Lines 194-195: "...because the initial coolant temperature is higher in the hotter seasons."

Fig 2: How is this graph determined? We don't know what the ambient temperatures were in each of the eight tests (two tests per day, four seasons)

Thank you for pointing out the lack of information. The detail of temperatures and plots were added in the caption of Fig.2 as following.

"Each plot was obtained from AM and PM tests conducted in four seasons (the experiments were conducted for a period of 5 days in each season). Average ambient temperatures were determined using the average value of the temperatures recorded during each driving test."

Fig 3: How many dyno tests were done? Were the ambient conditions of the PEMS test replicated here?

One chassis dynamometer test was conducted before the PEMS experiment in four seasons, so totally four dyno tests were conducted. The ambient conditions were almost the same between all the tested environments. We added this information in Lines 109-110. Lines 109-110: "n all the laboratory measurements, room temperature was set to be approximately 25 °C."

Fig 4: Why is the area of the EGR + SCR plots (third column) larger than the No EGR + SCR and EGR only plots? They should all occupy the same area because vehicle speed and driving force doesn't change across the three columns

Driving force and vehicle speed are different between three phases. No EGR+SCR results were obtained from first 10 min after the driving started. On the other hand,

EGR+SCR results were obtained from 30 min after the driving started. The driving root after 30 min included high speed and road gradient environments, and therefore, EGR+SCR results holds wide range of driving and speed compared with other phases.

Fig 5: As Fig 4 Fig 6: The R2 value for these graphs might be high, but there is large variation about the 1:1 line

The authors added the following sentence to refer the dispersion of predicted amount of exhaust emissions as for the reviewer's opinion.

Lines 277-284: "The dispersion of the plots might indicate the limitation associated with the application of the transient emission table in the estimation of NOx and CO2, as well as other presumable vehicular exhaust emissions. The dispersion resulted from the fact that the transient emission table modelled in this study simulated real-time exhaust emissions based on driving force and vehicle speed. Real-time prediction is associated with several uncertainties that cannot be taken into account by the two parameters, driving force and vehicle speed, such as transient high acceleration, emission control systems settings by the manufacturer, etc. Nevertheless, the predicted results shown in Figure 6(a) and (b) include high linearity even for the real-time measurement results. The results also highlight the possibility of being applied in emission inventory evaluation in future."

Please also note the supplement to this comment:
https://amt.copernicus.org/preprints/amt-2020-286/amt-2020-286-AC2-supplement.pdf

---

## Author Comment (AC3) · 29 Dec 2020

The authors are grateful to the referee 2, Mr. Nikiforos Zacharof, for giving us the critical insights and suggestions. The manuscript has been carefully modified based on the opinions. Sometimes there were difficult parts to be modified and the authors explained why we did not make change. The modified sentences based on the referee 2's suggestions are highlighted in green.

1 General comments 1.1 Introduction The authors should pay more attention to the introduction and make clear what is the purpose of the paper. There are several statements about the representativeness of the laboratory testing compared to the on-road

measurements, but it is not clear why this is discussed, while both approaches have pros and cons. Without proper listing of them cannot enable a comparison between the two approaches. In this section, it is needed to clarify better the goals of the paper and why the authors conducted this research.

Thank you for the suggestion. According to the opinion, the sentence was modified to promote the 'purpose' of this study to be easily understandable for the readers. The final goal of this study has already been included in the final sentence of Introduction.

Lines 75-78: "The purpose of this study is two-fold. First, chassis-dynamometer-based and RDE measurements using PEMS were conducted on a heavy-duty Japanese vehicle to determine the importance of RDE specific factors, including the ambient temperature and road gradient, among others. Second, the obtained experimental results were analysed based on two parameters, i.e., the driving force and vehicle speed, . . ."

1.2 Methodology Please describe the methodological approach (testing, analyses, etc) by offering an overview before moving into describing the testing process. With this structure, the reader is confused as the text moves directly into explaining the testing, but there has been no frame to put it into. Also, a clear description of the post-processing and the analyses that are performed to reach the goal of the paper is missing. Please provide an overview of the approach, describe each step of the approach and indicate all the analyses that you've performed. Additionally, please avoid presenting equations that a reader would already be familiar to, e.g. formula of acceleration.

The authors convinced for the opinion and following sentences were added in the revised manuscript. The redundant formula in equation (2), dv/dt, was removed according to the suggestion.

Lines 85-89: "Three methods were used in this study: laboratory test using the chassis dynamometer, RDE measurements using PEMS, and data analysis to construct the estimation model for the emissions inventory. For a detailed analysis of the experimental results from the RDE measurements, data processing methods included data

smoothing for the high-resolution time profiles of the emissions data and the extraction of the road gradient from official open sources. Details on these methods are described in sections 2.1 to 2.3."

1.3 Results and discussion The authors need to improve the presentation of the results in terms of text structure and language. Several times it has been unclear what they wanted to say, while some other times the text was repetitive. Please present all the analyses that help you answer those questions that you have set. At this point, it seems only a simple presentation of the measurement results. This is useful to enhance the literature in the topic, but it does not add up significantly, while there is potential for this. Please consider adding a more detailed sensitivity analysis, an energy analysis (energy share on wheel, auxiliaries, etc) and an improved comparison between the WHVC and on-road conditions. These are just recommendations to improve the paper, based on what is already included. Consider expanding the analyses as you see fit.

Thank you for the suggestions. The structures of the discussion were completely revised based on the three reviewer's opinions. The authors treated sensitivity analysis section 3.4.2 as the complement explanation to the transient table analysis to clarify what factors are important to the exhaust emissions. Hence, if it is possible, we do not want to further discuss the sensitivity topic in this manuscript (further, enough measurement data to be used in the sensitivity analysis in the suggestion were not collected). The next PEMS experiment is now ongoing, and the authors are intending to take detail sensitivity analysis including energy share or the effect of vehicle parts into consideration. Again, we are grateful for the valuable advices.

1.4 Conclusions The conclusion sector is unclear to what outcome the analysis reached. This section should avoid repeating what has already been presented in the discussion and actually reach to some conclusions and evaluate whether the paper's goals have been achieved or not, what new questions have arisen and what could be done next. The sole outcome is that the road gradient is the most important factor affecting NOx and CO2 emissions, which is hardly new. The authors conducted an

experimental campaign that should gain visibility, but it needs more analyses, reach some solid conclusions that could actually contribute in the field.

Conclusion sometimes repeats the outcome described in the previous sections to highlight the important topics in this study, especially base experimental results and transient table analysis. The finding of the effect of road gradient is not new but quite important in this study, so the authors decided to retain it. The novelty of this study was added in the Introduction as the following sentence.

Lines 80-81: "To the best of our knowledge, this is the first study in which PEMS measurement results are applied in the development of an estimation model for vehicular exhaust.

Based on the suggestion, to make clear the final goal of this research with what has still not been unsolved, following sentences were added in the revised manuscript."

Lines 327-331: "The experiment as well as the detailed analysis were conducted only for one heavy-duty vehicle in the Japanese market. In future, it is expected that further studies would be conducted to obtain the variability of real-world vehicular exhaust emissions; after the measurements results for the consistent number of vehicles have been obtained, the emissions inventory based on real-world measurements should be evaluated for use in policy making regarding air quality treatments."

2 Specific comments Line 20 Please provide references and/or examples of major countries/regions that are concerned with pollutants.

According to the suggestion, related references were added with following sentence.

Lines 22-23: "...including the United States, the European Union, China, India, and Japan (Akimoto et al., 2015; Costa et al., 2014; Ravindra et al., 2016; Sullivan et al., 2018; Yang and Wang, 2017)."

Line 23 Please provide reference to the NASA report or database.

Thank you for the clarification. The exact reference has been provided by the webpage as following. National Aeronautics and Space Administration Webpage.: https://climate.nasa.gov/vital-signs/global-temperature/ (accessed on December 15, 2020).

Line 25 The Saito 2010 reference is already 10 years old. Please add also a projection from newer studies if possible.

The authors are convinced about the opinion. Following recent reference from IPCC (2018) was added in the manuscript.

Summary for Policymakers. In: Global Warming of 1.5°C. An IPCC Special Report on the impacts of global warming of 1.5°C above pre-industrial levels and related global greenhouse gas emission pathways, in the context of strengthening the global response to the threat of climate change, sustainable development, and efforts to erad-icate poverty.

https://www.ipcc.ch/site/assets/uploads/sites/2/2019/05/SR15_SPM_version_report_LR.pdf (accessed on November 24, 2020).

Line 25 – 26 "... photochemical oxidant is a well known short-term air pollutant ..." It is not clear which photochemical oxidant the authors refer to and also there's lack of any reference to back this statement. Please elaborate on the whole sentence.

Thank you for pointing out the lack of information. The following sentence was added with the citation of Sillman (1999).

Line 29: "...photochemical oxidants, mainly composed of ozone,..."

Sillman S. The relation between ozone, NOx and hydrocarbons in urban and polluted rural environments. Atmos. Environ., 33, 1821-1845, https://doi.org/10.1016/S1352-2310(98)00345-8, 1999.

Line 30 – 31 "To address the problems of global warming and photochemical pollutants,

it is necessary to mitigate air pollution." This statement is not quite accurate. Pollutants cause health problems mainly and also sometimes have a global warming potential. The major factor for global warming from the automotive sector are CO2 emissions. CO2 is not a pollutant and it doesn't cause health problems by inhaling it. Please elaborate.

Thank you for pointing out the ambiguous expression. CO2 is not a health-hazardous component but despite this fact, CO2 is categorized as long-term air 'pollutant' when it is emitted by the anthropogenic sources such as vehicles, power plants etc[1]. Therefore, the sentence was modified as following.

Line 35: "...to mitigate the emissions of long- and short-term air pollutants."

[1]https://www.nationalgeographic.com/environment/global-warming/pollution/#:∼:text=Though%20many%20living%20things%20emit,as%20gasoline%20and%20natural%20gas.

Line 40-41 "In general, the laboratory temperature is set at approximately 25 °C, and it cannot be easily changed via the normal laboratory system." This is not quite true, the temperature in the laboratory and more specifically in the vehicle test cell can be adjusted to a range of different temperatures. In some cases, however, it could require additional investment in infrastructure such as to achieve a temperature of -7 °C that are required in some countries. The 25°C temperature is mandated by the testing protocol and the regulation that has adopted this protocol and not necessarily from the technical capabilities of the laboratory. Please elaborate the sentence.

Thank you for the detail clarification. The previous sentences indicated the difficulty of arranging arbitrary room temperature because of the exhaust heat from the laboratory system including the heat from vehicular exhaust. The sentences were modified as following.

Lines 45-46: "...it cannot be easily set to the arbitrary temperature (especially a low temperature range) owing to the exhaust heat from vehicle and measurement ma-

chines via the normal laboratory system. . .”

Line 43 “. . . a specific activation temperature that cannot be attained in cold seasons
. . .”, this is quite a bold statement and not entirely true. The activation temperature will
be reached at some point, but under cold conditions, it could take longer. In the case
of small trips and low temperatures, then it is possible that the activation temperature
is not reached. Please elaborate.

Thank you for the clarification, this sentence was also pointed out by another reviewer.
The relationship of ambient temp. and pollutant's emission is described in the results
section, so in the introduction, the sentence was modified as following.

Lines 47-50: “It has been noted that environmental temperature considerably influ-
ences the amount of exhaust emissions (detailed explanations regarding this obser-
vation are provided in section 3.1), leading to the release of a large amount of pollu-
tants (including NOx) into the atmosphere in low ambient temperature conditions (the
MOVES2010 Report by the U.S. Environmental Protection Agency, 2010).”

Line 44 – 46 “Moreover, the road gradient also influences the amountof exhaust emis-
sions because it directly affects the driving force, which is presumed to be proportional
to CO2 and other exhaust emissions.” This statement is not correct. The gradient in-
creases the required engine load and it indeed increases CO2 emissions, but this is
not done necessarily linearly. The road gradient force formula is defined as m*g*sin(a)
(m = vehicle mass, g = acceleration of gravity, a = road grade). The engine needs to
operate at a higher load to compensate for this force, but the operation point depends
also on the transmission ratio. In any case, the road grade increases fuel consumption
and therefore CO2 emissions – not necessarily linearly. In the case of pollutants, they
could increase but this depends on the engine operation point that affects what kind of
pollutants are produced in the engine. Whether these pollutants make it to the exhaust
relies heavily on the operation of the aftertreatment systems.

Thank you for the clarification. The authors also agree that road gradient does not

linearly affect to the driving force, and it depends on the parameters of engine torque and transmission ratio. In the analysis of this study on Fig.4 and Fig.5, the emission tables were evaluated treating with the two variables: driving force and vehicle speed. Vehicle speed was chosen to exhibit the engine rotation factor which corresponds to the transmission ratio, for example, in the high transmission range, engine rotation would increase. After-treatment systems introduced in the tested vehicle were EGR and urea-SCR. Fig.4 and Fig.5 treated those two after-treatment contribution by separating the effect of after-treatment tools in terms of operation time of the tools. Meanwhile, the term 'proportional' was modified by "...have a negative effect on...".

Line 46 "Consequently ..." This statement does not exactly explains why the governments are making PEMS tests and it seems like a leap of thought. Also no references are provided to back this statement. The following statement in line 48 "Gallus et al. (2017) ..." refers to an experimental campaign and not to the adoption of PEMS as an official protocol to define emissions. The following references also refer to research campaigns and not protocols. Please re-phrase and elaborate.

According to the suggestion, the following sentence with several references related to the political issues of RDE was added.

Lines 54-59: "For this reason, the European Union has implemented a regulation for RDE from light-duty vehicles using portable emissions measurement systems (PEMS) (Valverde et al., 2020). Consequently, countries, such as the United States, China, and India, have made decisions to implement RDE measurements as a regulatory test, and Japan is now considering the implementation of regulations via real-world measurements using PEMS (Giechaskiel et al., 2019). Currently, in terms of the research field of atmospheric science, many researchers have conducted RDE measurements using PEMS."

Line 55 "Nevertheless, the conduction of road measurement experiments using PEMS is a relatively new domain, and only a few studies have been performed to assess the

analytical data . . ." This is not quite true, please check the literature on this issue an elaborate. PEMS is common in testing, especially on light-duty vehicles, where in the European Union at least is part of the vehicle type approval procedure for pollutants requires Real Driving Emission (RDE) testing (Regulation (EU) 2017/1151). The cold start is important for light-duty vehicles as they are often doing short trips and the aftertreatment systems do not reach always optimal operation temperature. However, for heavy-duty vehicles the cold start effect is limited as the vehicles operate for long time (e.g. 8 hours or longer for a typical city bus operation) and they are compensated by the long operation time.

Thank you for the clarification. RDE by PEMS has been adopted as the type approval test in 2017, which is relatively (or quite) newer than chassis dynamometer test. We added "compared with chassis dynamometer experiments" in lines 67-68.

The driving duration of heavy-duty vehicles depends on the use of the vehicle. City bus could be driven continuously for a long duration but the vehicle for house-move could be sometimes driven with stop and run. The emission from short-time parking is sometimes equivalent to the cold-start emission especially in the cold season. Further, heavy-duty vehicles will not be substituted by the zero-emission vehicles in the near future because of the difficulty of high capacity battery. For those reasons, the RDE experiment for heavy-duty vehicle is also important to mitigate both current and future air quality issues.

Line 60 The term "classical mechanics" is a bit redundant in this context. Please consider to elaborate.

According to the suggestion, the words "classical mechanics" were modified as a following sentence.

Lines 77-78: "Second, the obtained experimental results were analysed based on two parameters, i.e., the driving force and vehicle speed, to develop. . ."

Line 62 Please take into consideration the instrument accuracy. PEMS can be utilized everywhere on-road, but could face accuracy issues, while these problems are minimized in the laboratory where the methodology and the instruments (e.g. bag result analysis) could be more accurate.

According to the suggestion, the sentence regarding the accuracy of PEMS was added with the citation of the study from Cao et al (2016).

Lines 65-69: "The accuracy of PEMS is also important in discussing RDE test results. Cao et al. (2016) conducted a study to clarify the accuracy of gaseous pollutants measured by PEMS, concluding that NOx emissions would sometimes be overestimated in the low NOx concentration range, accounting for a $\sim$50 % maximum from the reference NOx concentration owing to analyser drift. These results suggest the limitation of PEMS accuracy when the measurement system is used in the type approval test."

Line 98 The Eq. 1 seems redundant, but it could be retained. However, what the authors are describing here is a rolling average with a step of 5 observations. Please clarify.

Thank you for the suggestion. The detail explanation of eq.1 was added after eq.1 as the following sentences.

Lines 129-133: "In equation (1), vehicle speed was smoothed using the 5-point average of the speeds at neighbouring time steps. We determined the averaging number of vehicle speed as 5 points based on two concepts. First, the averaging number of vehicle speed should be minimized as much as possible to maintain the high resolution time step. Second, the dispersion should be sufficiently lower than the measured vehicle speed. It is also worth noting that the averaging number of vehicle speed may depend on the measurement tools, and the 5-point value was suitable in this study."

Line 101 The Eq. 2 seems redundant as it presents the calculation of acceleration based on speed, which must be a well-known topic for the reader. Retain it if you

consider it useful for your narration, otherwise please remove and elaborate.

The authors thought that eq.2 is important to show the concrete definition of central differential method although it is a well-known mathematical method. But we also thought that the definition of acceleration, a=dv/dt, is actually redundant and this definition was deleted from eq.2.

Line 103 The authors in this section (2.3.2) describe the synchronization of the data. For every data that are retrieved from the vehicle there's a time lag between the phenomenon that occurs and the measurement time. Combustion process and exhaust emissions are directly correlated as the latter are products of the former process but the data are retrieved with a time lag as gases need to travel from the combustion chamber to the exhaust where they are measured. Data timestamps from the combustion chamber and the exhaust would correspond to different events and they need to be synchronized. This applies to all the sensors but not all the events can be correlated in this way. This section seems that it is not needed as this process is standard for the post-processing. However, if you retain it please elaborate this section and consider merging it with another one where you describe the data post-processing.

Actually the data synchronization is required for vehicle exhaust measurements when the detail analysis is conducted. The synchronization method could be different among the researchers or engineers, and in this study, the optimization by the cross-correlation function was chosen. It is important to show the definition of synchronization method since the method is not widely known by the researchers or engineers in this field. The sentences were retained but the section 2.3.2 was merged to 2.3.1 based on the suggestion.

Line 120 The $\mu$r and $\mu$a are not defined. Please define that are the rolling resistance and air drag coefficients.

Those two parameters were defined as "rotation friction coefficient" and "air friction coefficient" in lines 141-142, so the authors would like you to check it again.

Line 122 You mentioned the vehicle weight was set at 5880. Have you measured the vehicle on a balance or did you derive this value from your calculations?

Vehicle and PEMS weights are catalog value. Battery weights were measured. The following explanations were added in the manuscript.

Lines 156-158: "The total vehicle weight, m, including the weight of the vehicle itself (4,920 kg), cargo, such as PEMS (≒ 200 kg), four batteries for PEMS (37 kg × 4 ≒ 150 kg), driver and operator (55 kg × 2 ≒ 110 kg), and other measurement-related parts (500 kg), was set as 5880 kg"

Line 142 The "engine room" could mean the whole compartment where the engine is placed. Please consider replacing it with the more appropriate term "combustion chamber".

According to the suggestion, "engine room" was replaced by "combustion chamber".

Line 186 "which is not taken into account by the WHVC approach.". The WHVC development was based on real-world data in order to produce a representative situation of real-world conditions. The claim that the testing conditions are not represented in the WHVC is useful as observation, but it needs to back it with enough data. First, it should be quantified how many times are encountered the testing conditions that were outside the WHVC approach. The following questions must be answered. Has it been on every test, has it been affected by the driving style or has it been due to the requirements of your experiment? Second, it is needed to compare to real-world route conditions to the overall WHCV approach and quantify the effect and its significance.

Thank you for the clarifications. The authors were convinced about the opinions and following sentences were added in the revised manuscript.

Line 227: "…were observed in all RDE test conditions."

Lines 228-231: "We note that the acceleration distribution also depends on the driving feature, and the distribution changes for each driver (Ericsson, 2001). The drivers of

the RDE and chassis dynamometer experiments were different in this study, such that the results of the difference in the acceleration distribution can also be attributed to the difference in the driving feature."

As described in the manuscript, the route of RDE in this study did not include high speed range which is included in WHVC. For this reason, the authors did not compare the detail of PEMS results and WHVC results. PEMS measurement with high-speed range in the motor-way RDE measurements are now ongoing, and the results will be opened in the coming year. Thank you for your reconsideration.

Line 199-207 This is methodology, please move to the respective part.

Thank you for the suggestion. After the careful consideration based on the suggestion, we thought that it is better to retain the sentences at this part. The method to obtain transient map from driving force, vehicle speed, and NOx and $CO_2$ emissions has already been described in the Methodology section. On the other hand, the background of the transient map is described at section 3.4.1. The background could also be described in Introduction, but we thought that it is more reader-friendly to remain these sentences (Lines 243-247) at the top of section 3.4.1. Please reconsider our decision.

Line 217 - 218 It should be clarified that the use of air conditioning poses an additional load to the engine that could make the engine operate in less optimum operation ranges under some conditions. In this way it could lead to an increase in pollutants.

Thank you for the suggestion. Based on the suggestion following sentence was added in the revised manuscript.

Lines 264-265: "The use of the air conditioner leads to an additional load on the engine that may render engine operation at less optimum operational ranges, leading to an increase in emissions."

Line 245 – 250 It is stated several times the that "a parameter depends on the same parameter" such as "the road gradient depends on the cosine of the road gradient",

which is quite obvious. Please elaborate the whole text and remove such statements.

Thank you for the clarification. The term was modified from "road gradient" to "road angle". It seems there is no similar miss-sentence.

Line 253 – 254 "the WHVC driving mode is currently applied worldwide." I am not sure whether this is true and it is quite vague. Please mention major countries/regions that use this protocol and for what reasons. In Europe for instance, the heavy-duty vehicle type approval procedure is performed through a simulatory approach that utilizes other driving cycles.

WHVC is mainly used for the research purposes to measure and compare exhaust emissions for various heavy-duty vehicles (although WHVC is also required for the imported vehicles from abroad which could not be tested by engine bench.). The type approval tests, on the other hand, is composed of engine bench tests, WHSC and WHTC, in Europe, North America, Japan, and Australia (https://dieselnet.com/standards/cycles/index.php). To account for these issues, following sentence was added in the revised manuscript.

Lines 309-310: "...including Europe, the United States, and Japan, among others, mainly for research purposes (DieselNet Webpage)."

3 Technical corrections Line 28 "(O'Neill et al., 2004; Chappelka and Samuelson, 1998; Wang et al., 2017)" please re-arrange in chronological order.

Thank you for pointing out the order of citations. The order was modified chronologically.

Line 44 Please provide an accurate reference for the EPA.

The citation was modified to be "Report of MOVES2010 by U.S. Environmental Protection Agency, 2010". The exact reference was described as following.

U.S. Environmental Protection Agency: MOVES2010: Highway Vehicle Temperature, Humidity, Air Conditioning, and Inspection and Maintenance Adjustments. https://www.google.co.jp/url?sa=t&rct=j&q=&esrc=s&source=web&cd=1&ved=2ahUKEwjZrOKxvOzoAhWB3mEKHSJWCh hMuLXe2RqyY5868 (accessed on January 16, 2020).

Line 54 "(Kousoulidou et al, 2013; Kwon et al, 2017; Liu et al, 2009; Luján et al, 2018; O'Driscoll et al, 2016)" please re-arrange in chronological order.

Thank you for pointing out the order of citations. The order was modified chronologically.

Line 70 "current Japanese regulation set in 2016." Please cite the exact law. Line 76 "... in our previous work..." Please avoid using possessive pronouns and especially in the first person. Consider removing them entirely and retain the reference to your work. If you want to retain the connection with your work for any reason, please consider using the third person e.g. "in the authors' previous work".

The current regulation for heavy-duty vehicle was added in Table S1 of supplementary information. The possessive expression of the citation was removed to "in the previous study". Thank you for the clarification.

Line 259 "in the Japanese market were conducted", I think you mean a vehicle that is available in the Japanese market, but it unclear. Please correct.

The term 'used' was added before the words "in the Japanese market were conducted".

Please also note the supplement to this comment: https://amt.copernicus.org/preprints/amt-2020-286/amt-2020-286-AC3-supplement.pdf

---

## Author Response (AR2)

**For handling editor**

*In addition to the points raised by the referee, it would be good to carefully check the use of the language in the paper. After that, the paper can be published.*

The authors are grateful to the handling editor to consider our manuscript as publication. Manuscript has been carefully checked by all co-authors with the improvements pointed out by the referee. I would like the editor to confirm that corresponding author will change the affiliation from coming April 2021 and thus, two contact e-mail addresses (before and after April) are noted in the correspondence part. Thank you for the consideration.

**For Referee**

*The paper needs some minor corrections in terms of language and technical terms. Please go through the document and elaborate the text. Other than that it was good to see that various issues have been addressed. The paper describes in detail the experimental process and highlights several issues that could be useful in the field of heavy-duty vehicle testing. This especially useful as various regions in the world are setting respective emission targets. One important aspect of the study is the divergence of the driving profile from the WHVC, which could be a basis for future research.*

The authors are grateful to the referees to check manuscript as well as suggesting us the valuable modifications. The minor corrections were taken into account based on the following opinions.

*There is a series of mostly minor issues to be corrected:*
*Line 16: "The constructed tables well replicated…", please elaborate sentence to "The constructed tables replicated well…"*

The order of words was improved based on the suggestion.

*Line 23 – 25: "Global warming… (Saito, 2010)." This sentence is quite long and a bit difficult to follow. Please elaborate and consider splitting it into two.*

Thank you for pointing out the difficulty of the sentence. The sentence was split into two sentences as following.
Lines 26-29: "Climate change continues to occur, with the global temperature increasing annually. It has been reported that …"

*Line 25: "photochemical oxidant", please consider using the term in plural or as "photochemical oxidant compounds".*

We modified the term as plural, "photochemical oxidants", thank you for pointing out mistake.

*Line 88: "two times" consider replacing it with "twice".*

According to the suggestion, the words "two times" were substituted by "twice".

*Line 94: "The vehicle acceleration (km/s2) was…", please check the units, there seems to be a mistake. I would expect to see m/s2.*

> The unit of acceleration data monitored by PEMS measurements was km/s2, on the other hand, the unit treated by data analysis was converted to m/s2. Each unit is defined in both lines 124 and 151. The conversion method is relevant (just multiplying 1000 to km/s2), so this is not mentioned in the manuscript.

*Line 142: Please correct the term "inside the engine room" with a more suitable term such as "combustion chamber", otherwise it conveys the wrong message. Apply this to all other relative cases.*

> Thank you for the suggestion. All terms were modified.

Line 176: "the engine is cooled", it is not clear whether the engine is cold or it is cooled down explicitly for the purposes of the experiment. Same applies in line 178 "vehicle body is also cooled".

> Thank you for the suggestion. The terms "over several parking durations" were added after "the engine is cooled" and "vehicle body is also cooled" to make the sentences clearer.

*Line 203: "detoxification system" is an uncommon term and it could be a bit misleading. Please change to after-treatment systems.*

> Thank you for clarification. All the terms were modified to "after-treatment systems".